# $f$-FERM: A Scalable Framework for Robust Fair Empirical Risk Minimization

**Sina Baharlouei** [*]      **Shivam Patel** [†]      **Meisam Razaviyayn** [*]

## Abstract

Training and deploying machine learning models that meet fairness criteria for protected groups are fundamental in modern artificial intelligence. While numerous constraints and regularization terms have been proposed in the literature to promote fairness in machine learning tasks, most of these approaches are not amenable to stochastic optimization due to the complex and nonlinear structure of constraints and regularizers. Here, the term "stochastic" refers to the ability of the algorithm to work with small mini-batches of data. Motivated by the limitation of existing literature, this paper presents a unified stochastic optimization framework for fair empirical risk minimization based on $f$-divergence measures ($f$-FERM). The proposed stochastic algorithm enjoys theoretical convergence guarantees. In addition, our experiments demonstrate the superiority of fairness-accuracy tradeoffs offered by $f$-FERM for almost all batch sizes (ranging from full-batch to batch size of one). Moreover, we show that our framework can be extended to the case where there is a distribution shift from training to the test data. Our extension is based on a distributionally robust optimization reformulation of $f$-FERM objective under $\ell_p$ norms as uncertainty sets. Again, in this distributionally robust setting, $f$-FERM not only enjoys theoretical convergence guarantees but also outperforms other baselines in the literature in the tasks involving distribution shifts. An efficient stochastic implementation of $f$-FERM is publicly available [1].

## 1 Introduction

Machine learning models are increasingly deployed in critical applications ranging from healthcare (Ahmad et al., 2018) to image processing (Krizhevsky et al., 2017), education to job recruitment (Boselli et al., 2018), and social networking to cybersecurity (Xin et al., 2018). Machine learning practitioners have adopted learning algorithms to fathom inherently difficult and crucial problems. However, naïve deployment of these models may lead to serious shortcomings such as biased predictions against minority groups (Angwin et al., 2016; Buolamwini & Gebru, 2018), vulnerability to adversarial attacks (Madry et al., 2017; Carlini & Wagner, 2017; Baharlouei et al., 2023), or lack of generalizability (Arjovsky et al., 2019). Consequently, it is of utmost importance to have reliable and trustworthy models that are, in particular, fair and comply with equality norms and provisions worldwide (Act, 1964; Elford, 2023).

With the increasing concern for the trustworthiness of unchecked machine learning algorithms, a broad class of paradigms has been proposed to counteract and mitigate both the cause and effects of model unreliability. Imposing statistical independence between model output and particular input features is of interest in various domains, especially when the generalization of a trained model is based on a collection of spurious features present in the training dataset (Dwork et al., 2012; Hardt et al., 2016; Yan et al., 2017). These could be sensitive features like gender, race, age, and/or income in the context of fairness or confounding factors like environmental artifacts in the context of image classification (Arjovsky et al., 2019). Existing literature on imposing statistical independence between selected input features and model outputs is directed into three approaches: pre-processing, post-processing, and in-processing methods.

Pre-processing methods entail upstream changes made in datasets to mask sensitive features or reduce the dependency of output variables on sensitive features through transforming data in a stage before the training phase (Kamiran & Calders, 2012; Zemel et al., 2013; Ustun et al., 2019). Post-processing methods involve model-specific adjustments to the model's output to ensure the independence of

---

[*]University of Southern California (`baharlou, razaviya@usc.edu`)

[†]Department of Electrical Engineering, IIT Bombay (`shivamapatel2002@gmail.com`)

[1]https://github.com/optimization-for-data-driven-science/f-FERM

predictions and sensitive attributes (Hardt et al., 2016; Alghamdi et al., 2022). While pre-processing and post-processing methods do not affect the training procedure, they fail to exploit underlying training mechanisms for the best achievable accuracy-fairness tradeoffs. Unsurprisingly enough, optimizing accuracy and fairness jointly (in-processing) leads to better tradeoffs than sequentially optimizing fairness and accuracy in a pre-processing or post-processing fashion.

In-processing methods alternatively add fairness constraints or regularizers, penalizing dependence between sensitive attributes and output variables. (Zafar et al., 2017) utilizes covariance as the measure of independence between the sensitive attributes and the predictions. While such a measure is amenable to stochastic updates, it fails to capture correlations beyond linear. Alternatively, several non-linear measures such as Rényi correlation (Baharlouei et al., 2020), $\chi^2$ divergence (Lowy et al., 2022), $L_\infty$ distance (Donini et al., 2018), and Maximum Mean Discrepancy (MMD) (Prost et al., 2019) are proposed in the literature to establish the independence of the predictors and sensitive attributes. In-processing techniques can be model-specific (Wan et al., 2021; Aghaei et al., 2019) or generalizable to different training algorithms (Baharlouei et al., 2020; Lowy et al., 2022).

In the spirit of in-processing methods, input data-driven constraints or regularization terms are used to modify training objectives of problems like learning generalizable models to new environments, invariant learning, and learning in the presence of distribution shifts (Arjovsky et al., 2019; Mary et al., 2019; Baharlouei et al., 2020). Such constrained/regularized reformulations are prevalent in learning robust classifiers against adversarial attacks (Sinha et al., 2018), meta-learning (Balaji et al., 2018), federated learning (Deng et al., 2023), and alternative learning paradigms such as learning distributionally robust optimization (DRO) models (Kuhn et al., 2019; Levy et al., 2020), tilted empirical risk minimization (TERM) (Li et al., 2020), and Squared-root Lasso (Belloni et al., 2011).

While in-processing techniques outperform pre-processing and post-processing approaches, they are not scalable to large datasets because of a lack of adaptability to stochastic optimization (Mary et al., 2019; Lowy et al., 2022). All aforementioned examples consist of regularization terms in their objective functions where the gradient cannot be described as a linear combination of data point functions. As a result, applying stochastic gradient descent or other stochastic first-order methods on the objective functions of such problems might not converge, especially for small batch sizes.

Motivated by this, (Lowy et al., 2022) proposes a stochastic optimization framework for Exponential Rényi Mutual Information as the measure of independence. More recently Zhong & Tandon (2023) use $f$-divergences as regularization terms to establish the independence between sensitive attributes and predictions. They estimate the $f$-divergence regularizers offline through multi-layer neural networks to avoid the computational challenges of devising scalable stochastic methods for nonconvex min-max problems. Our approach, on the other hand, directly solves the variational formulation for both full-batch and stochastic settings with convergence guarantees to non-spurious solutions. In Section 2, using the variational representation of $f$-divergences, we present a convergent stochastic optimization framework for fair learning via $f$-divergences. (Lowy et al., 2022) is a special case of $f$-divergences where $f(t) = t^2 - 1$ ($\chi^2$ divergence). Aside from $\chi^2$, all other divergences listed in Table 1 are not introduced in the literature to the best of our knowledge.

Designing convergent stochastic algorithms for fair empirical risk minimization can be further explored in scenarios involving changes in the data distribution from the source to the target domain. Detection and mitigation of biases against protected groups in the presence of distribution shifts have been extensively studied in recent years. Lechner et al. (2021) theoretically shows that learning fair representations (pre-processing) is nearly *impossible* for the popular notions of fairness, such as demographic parity in the presence of the distribution shift. Ding et al. (2021), on the other hand, experimentally demonstrates that applying post-processing fairness techniques (Hardt et al., 2016) to learn fair predictors of income concerning race, gender, and age fails to transfer from one US state (training domain) to another state. Overlooking distribution shifts can lead to catastrophic decisions threatening the well-being of human subjects when deploying a trained model in certain hospitals to other hospitals (Schrouff et al., 2022). The current literature for handling distribution shifts with in-processing methods relies on certain assumptions on the type of distribution shift (demographic shift (Fang et al., 2020; Du & Wu, 2021; Maity et al., 2021; Giguere et al., 2021), label shift (Dai & Brown, 2020), and/or covariate shift (Rezaei et al., 2021; Singh et al., 2021)) or explicit access to the **causal graph** (Mishler & Dalmasso, 2022; Schrouff et al., 2022) of predictors, sensitive attributes, and target variables. As a result, they face practical limitations and cannot cope with most real-world problems involving complex shifts that cannot be categorized in the ones assumed in their works.

Alternatively, Taskesen et al. (2020) provides convex objective functions for imposing fairness on logistic regression using constraint optimization. Staib & Jegelka (2019) use MMD for defining uncertainty sets around training distribution, whereas Husain (2020) use Integral Probability Measure (IPM) to mitigate the distribution shift. The main limitation of these approaches is their reliance on the convexity of the underlying learning model and lack of scalability due to incompatibility with stochastic optimization algorithms. Wang et al. (2023) uses the Maximum Mean Discrepancy (MMD) distance between the spectral norm of the Hessian matrix at advantaged and disadvantaged data points. However, they do not provide convergence guarantees for their proposed algorithm to any notion of optimality. In addition, the method is not necessarily amenable to stochastic updates. While we naturally define the uncertainty set directly on the joint distribution of sensitive attributes and predictions, they use the curvature of the obtained solution quantified by the norm of the Hessian matrix as a heuristic for promoting the robustness of the fair solution.

**Contributions:** This paper establishes a scalable (stochastic) fair empirical risk minimization framework through regularization via $f$-divergences ($f$-FERM) for both standard and distributed shift settings. $f$-FERM presents a unified methodology based on the Legendre-Fenchel transformation, enabling us to develop theoretically convergent first-order stochastic algorithms when only small batches of data are available at each iteration. Further, we have presented the first distributionally robust optimization framework under $\ell_p$ norms uncertainty sets covering nonconvex losses such as neural networks. The presented framework for fair inference in the presence of distribution shifts does not rely on the causal graph describing the causal interaction of input features, sensitive attributes, and target variables, which is rarely available in practical problems.

**Paper Organization:** We structure our response towards designing scalable, robust, and fair algorithms into two sections. Section 2 motivates the design of unbiased gradient estimators of objectives with information-theoretic $f$-divergence regularizers. In Section 3, we present our approach for fair inference in the presence of the distribution shift in detail. Our experiments provide an extensive examination of various $f$-divergences and their suitability as regularizers and also show the consistency of our method across all batch sizes in contrast to existing benchmarks. Similar experiments are carried out for robust training on varying amounts of distributional shifts in data.

## 2 FAIR EMPIRICAL RISK MINIMIZATION VIA $f$-DIVERGENCES

A widely studied problem in algorithmic fairness is promoting a notion of group fairness, such as demographic parity, equalized odds, equality of opportunity, or sufficiency through an in-processing method. For these notions, we aim to establish a [conditional] statistical independence between the predictions (e.g., the creditworthiness of the individual) and the sensitive attributes (e.g., gender, race). For simplicity of presentation, we formulate all problems under the demographic parity notion, which requires statistical independence between the prediction and the sensitive attribute. Without loss of generality, all formulations and methods are generalizable to other aforementioned notions of group fairness by considering conditional random variables (see Appendix A). A popular in-processing approach for training fair (classification) models under the demographic parity notion is to regularize the empirical risk minimization:

$$\min_{\boldsymbol{\theta}} \quad \frac{1}{n} \sum_{i=1}^{n} \ell(\hat{y}_{\boldsymbol{\theta}}(\mathbf{x}_i), y_i) + \lambda \mathcal{D}\Big(\mathbb{P}(\hat{y}_{\boldsymbol{\theta}}(\mathbf{x}), s), \mathbb{P}(\hat{y}_{\boldsymbol{\theta}}(\mathbf{x})) \otimes \mathbb{P}(s)\Big), \tag{1}$$

where $\boldsymbol{\theta}$ is the learning parameters (e.g., weights of the neural network); $\mathbf{x}_i \in \mathbb{R}^d$ is the $i$-th input feature vector; $y_i$ is the actual label/class for sample $i$; $\hat{y}_{\boldsymbol{\theta}}(\mathbf{x}_i)$ is the prediction of the model for sample $i$; and $\ell(\hat{y}_{\boldsymbol{\theta}}(\mathbf{x}_i), y_i)$ is the loss function measuring the "goodness-of-fit" for sample $i$. Here, $\mathcal{D}$ is a divergence between the joint probability distribution of the predictions and sensitive attributes and the Kronecker product of their marginal distributions. Recall that $\hat{y}_{\boldsymbol{\theta}}$ and $s$ are statistically independent iff $\mathbb{P}(\hat{y}_{\boldsymbol{\theta}}(\mathbf{x}), s)$ follows $\mathbb{P}(\hat{y}_{\boldsymbol{\theta}}(\mathbf{x})) \otimes \mathbb{P}(s)$. Thus, the second term in (1) is zero iff $\hat{y}_{\boldsymbol{\theta}}$ and $s$ are statistically independent (complete fairness under the demographic parity notion).

This section studies the fair empirical risk minimization regularized by a broad class of $f$-divergence measures. Let $\mathbb{P}$ and $\mathbb{Q}$ be two discrete probability measures taking values in $\mathcal{P} = \{1, \ldots, m\}$. The $f$-divergence between $\mathbb{P}$ and $\mathbb{Q}$ is defined as (Polyanskiy & Wu, 2022, Def 4.9)(see Appendix B for the general continuous case):

$$\mathcal{D}_f(\mathbb{P}, \mathbb{Q}) = \sum_{j=1}^{m} \mathbb{Q}_j f\left(\frac{\mathbb{P}_j}{\mathbb{Q}_j}\right) \tag{2}$$

The above definition, which is also known as $f$-mutual information (Lu et al., 2023; Csiszár, 1967), covers many known divergence measures used for imposing fairness, such as KL-divergence for the choice of $f(t) = t \log(t)$ (Shui et al., 2022), or $\chi^2$ divergence when $f(t) = (t-1)^2$ (Lowy et al., 2022). As shown in Appendix C, $\mathcal{D}_f$ in (1) is zero *if and only if* the probability distribution of $s$ and $\hat{y}_\theta$ *are statistically independent* for the choices of $f$ listed in Table 1. In addition, we prove that these $f$-divergences either cover or provide upper bounds for the popular notions of fairness violations in the literature, such as $\ell_p$ distances, Rényi correlation (Baharlouei et al., 2020), and demographic parity (equalized odds) violation. This means that by minimizing these regularizers, we are minimizing an upper bound of (other) popular fairness violation measures, and thus we are controlling them implicitly. Further, unlike Rényi correlation (Baharlouei et al., 2020; Grari et al., 2020), we can utilize Legendre-Fenchel duality (and variational representation) to develop (provably) convergent algorithms with **stochastic (mini-batch) updates**. This formulation and the resulting stochastic optimization algorithm are described in the next subsection.

## 2.1 A Convergent Stochastic Algorithm for fair ERM via $f$-Divergences

Let us start by rewriting (1) using $f$-divergences as the divergence measure:

$$\min_{\boldsymbol{\theta}} \quad \frac{1}{n} \sum_{i=1}^n \ell(\hat{y}_\theta(\mathbf{x}_i), y_i) + \lambda \sum_{\substack{j \in \mathcal{Y}, \\ k \in \mathcal{S}}} \mathbb{P}_s(s=k) \mathbb{P}_{\hat{y}_\theta}(\hat{y}_\theta) f\Big( \frac{\mathbb{P}_{\hat{y}_\theta, s}(\hat{y}_\theta = j, s = k)}{\mathbb{P}_{\hat{y}_\theta}(\hat{y}_\theta = j) \mathbb{P}_s(s = k)} \Big) \qquad (f\text{-FERM})$$

While the non-linearity of $f$-divergences in ($f$-FERM) empowers the underlying model to capture more complex dependencies between sensitive attributes and predictions compared to the linear measures (Zafar et al., 2017), the objective function can no longer be represented as a summation of functions over input data points. Consequently, one cannot directly apply the stochastic gradient descent method (or its variations, such as Adam) to the objective function in ($f$-FERM). In particular, directly evaluating the gradient of the objective function of ($f$-FERM) on a mini-batch of data leads to a statistically biased estimation of the entire objective's gradient. Such statistical biases prevent the convergence of algorithms such as SGD (even with a strongly convex minimization landscape) (Ajalloeian & Stich, 2020; Chen & Luss, 2018), let aside the more complex objectives arising in modern-day neural networks.

To derive stochastic algorithms, one can use the variational forms of $f$-divergences to delineate them as a pointwise supremum of affine transformation over probability densities. The most commonly used and well-behaved transform is the Legendre-Fenchel transform (often called the convex conjugates), which linearizes the dependence of the objective function to input data points using a variational reformulation. Particularly, we can rewrite ($f$-FERM) using the following result:

**Proposition 2.1.** *Let $f(\cdot)$ be a convex function. Then, ($f$-FERM) can be reformulated as:*

$$\min_{\boldsymbol{\theta}} \max_{A} \quad \sum_{i=1}^n \ell(\hat{y}_\theta(\mathbf{x}_i), y_i) + \lambda \sum_{\substack{j \in \mathcal{Y}, \\ k \in \mathcal{S}}} \Big[ A_{jk} \mathbb{P}_{\hat{y}, s}(\hat{y}_\theta = j, s = k) - f^*(A_{jk}) \mathbb{P}_{\hat{y}}(\hat{y}_\theta = j) \mathbb{P}_s(s = k) \Big] \quad (3)$$

*where $f^*(z) = \sup_{w \in \mathrm{dom}(f)} w^T z - f(w)$ is the Legendre-Fenchel transformation of the function $f$.*

*Proof.* The proof is standard and appears in Appendix D. $\qquad\square$

In order to solve (3), we will use (stochastic) first-order methods. Notice that $\mathbb{P}_s(s = k)$ is constant through the optimization procedure and is computed once by counting the number of data points whose sensitive attribute takes the value of $k$: $\pi_k := \mathbb{P}_s(s = k) = \frac{1}{n} \sum_{i=1}^n \mathbb{1}(s_i = k)$. Assume we use the softmax layer to compute the probabilities of different classes in our classification task (as it is standard in logistic regression or using neural networks for classification). Let $F_j(\mathbf{x}_i; \boldsymbol{\theta})$ be the $j$-th entry of the softmax layer output for datapoint $\mathbf{x}_i$, predicting the probability of class $j$. Then it is easy to show that we can obtain unbiased estimators of $\mathbb{P}_{\hat{y}_\theta}(\hat{y}_\theta = j)$ and $\mathbb{P}_{\hat{y}_\theta, s}(\hat{y}_\theta = j, s = k)$ using i.i.d. mini-batch $\mathcal{B}$ of data points. More precisely, we have

$$\mathbb{P}_{\hat{y}_\theta}(\hat{y}_\theta = j) = \frac{1}{n} \sum_{i=1}^n F_j(\mathbf{x}_i; \boldsymbol{\theta}) = \mathbb{E}\Big[ \underbrace{\frac{1}{|\mathcal{B}|} \sum_{i=1}^{|\mathcal{B}|} F_j(\mathbf{x}_i; \boldsymbol{\theta})}_{\hat{\mathbb{P}}_{\hat{y}_\theta}(j; \mathcal{B})} \Big]$$

$$\mathbb{P}_{\hat{y}_\theta, s}(\hat{y}_\theta = j, s = k) = \frac{1}{n} \sum_{i=1}^n F_j(\mathbf{x}_i; \boldsymbol{\theta}) \mathbb{1}(s_i = k) = \mathbb{E}\Big[ \underbrace{\frac{1}{|\mathcal{B}|} \sum_{i=1}^{|\mathcal{B}|} F_j(\mathbf{x}_i; \boldsymbol{\theta}) \mathbb{1}(s_i = k)}_{\hat{\mathbb{P}}_{\hat{y}_\theta, s}(j, k; \mathcal{B})} \Big].$$

$$(4)$$

Table 1: Unbiased Estimators for $f$-divergence Regularizers

| Divergence | $f(t)$ | The term $r_{jk}$ inside regularizer $\lambda \sum_{j,k} r_{jk}$ in (5) |
|---|---|---|
| $\chi^2$ | $(t-1)^2$ | $\pi_k[A_{jk}\mathbb{P}_{\hat{\mathbf{y}}_{\boldsymbol{\theta}}\mid\mathbf{s}_k} - (A_{jk} + \frac{\mathbf{A}_{jk}^2}{4})\mathbb{P}_{\hat{\mathbf{y}}_{\boldsymbol{\theta}}}]$ |
| Reverse KL | $-\ln t$ | $\pi_k[A_{jk}\mathbb{P}_{\hat{\mathbf{y}}_{\boldsymbol{\theta}}\mid\mathbf{s}_k} + (1 + \ln(-A_{jk}))\mathbb{P}_{\hat{\mathbf{y}}_{\boldsymbol{\theta}}}]$ |
| Total Variational | $\frac{1}{2}\lvert t-1\rvert$ | $\pi_k A_{jk}[\mathbb{P}_{\hat{\mathbf{y}}_{\boldsymbol{\theta}}\mid\mathbf{s}_k} - \mathbb{P}_{\hat{\mathbf{y}}_{\boldsymbol{\theta}}}]\mathbb{I}_{\{\lvert A_{jk}\rvert<1/2\}}$ |
| KL | $t\ln t$ | $\pi_k[A_{jk}\mathbb{P}_{\hat{\mathbf{y}}_{\boldsymbol{\theta}}\mid\mathbf{s}_k} - e^{A_{jk}-1}\mathbb{P}_{\hat{\mathbf{y}}_{\boldsymbol{\theta}}}]$ |
| Jensen-Shannon | $-(t+1)\ln(\frac{t+1}{2}) + t\ln t$ | $\pi_k[A_{jk}\mathbb{P}_{\hat{\mathbf{y}}_{\boldsymbol{\theta}}\mid\mathbf{s}_k} + \ln(2 - e^{A_{jk}})\mathbb{P}_{\hat{\mathbf{y}}_{\boldsymbol{\theta}}}]$ |
| Squared Hellinger | $(\sqrt{t}-1)^2$ | $\pi_k[A_{jk}\mathbb{P}_{\hat{\mathbf{y}}_{\boldsymbol{\theta}}\mid\mathbf{s}_k} + (A_{jk}^{-1} + 2)\mathbb{P}_{\hat{\mathbf{y}}_{\boldsymbol{\theta}}}]$ |

As a result, Problem (3) can be written as a linearly separable function of input data points ($\mathbf{x}_i$'s):

$$\min_{\boldsymbol{\theta}} \max_{A} \quad \frac{1}{n}\sum_{i=1}^{n}\left[\ell(\hat{y}_{\boldsymbol{\theta}}(\mathbf{x}_i), y_i) + \lambda \sum_{\substack{j\in\mathcal{Y},\\k\in\mathcal{S}}}\left[A_{jk}F_j(\mathbf{x}_i;\boldsymbol{\theta})\mathbb{1}(s_i=k) - f^*(A_{jk})\pi_k F_j(\mathbf{x}_i;\boldsymbol{\theta})\right]\right] \quad (5)$$

Thus, evaluating the gradient of the objective function w.r.t. the variables $\boldsymbol{\theta}$ and $\mathbf{A}$ over a random batch of data points leads to an unbiased estimator of the gradient of the objective function.

In addition to providing an unbiased estimator of gradients, the reformulation (5) has another crucial property: *the objective function is concave in* $\mathbf{A}$. Therefore, optimization problem (5) falls under the category of nonconvex-concave min-max optimization problems. That is, the objective is (possibly) nonconvex in $\boldsymbol{\theta}$ and is concave in $\mathbf{A}$. Thus, we can borrow tools from the (stochastic) nonconvex-concave min-max optimization literature (Lin et al., 2020; Razaviyayn et al., 2020a; Li et al., 2023) to derive a convergent first-order stochastic algorithm as presented in Algorithm 1. We listed the closed-form of $f(\cdot)$, $f^*(\cdot)$, for several widely-used $f$-divergence measures including KL-divergence, Reverse KL-divergence, $\chi^2$-divergence, Squared Hellinger distance, Jensen-Shannon divergence, and total variation distance in Table 1. For the derivation, see Appendix E.

---

**Algorithm 1** Stochastic Gradient Descent-Ascent (SGDA) for $f$-FERM

1: **Input**: $\boldsymbol{\theta}^0 \in \mathbb{R}^{d_{\theta}}$, step-sizes $\eta_{\boldsymbol{\theta}}, \eta_{\alpha}$, fairness parameter $\lambda \geq 0$, iteration number $T$, Batchsize $b$
2: **for** $t = 1, \ldots, T$ **do**
3: $\quad$ Sample minibatch of data $\mathcal{B}_t = \{(\mathbf{x}_{t1}, \mathbf{y}_{t1}), \cdots, (\mathbf{x}_{tb}, \mathbf{y}_{tb})\}$
4: $\quad \boldsymbol{\theta}^t = \boldsymbol{\theta}^{t-1} - \frac{\eta_{\boldsymbol{\theta}}}{b}\sum \nabla_{\boldsymbol{\theta}}\ell(\hat{y}_{\boldsymbol{\theta}}(\mathbf{x}), y) - \eta_{\boldsymbol{\theta}}\lambda\nabla_{\boldsymbol{\theta}}\left(A_{jk}^{t-1}\hat{\mathbb{P}}_{\hat{\mathbf{y}}_{\boldsymbol{\theta}},\mathbf{s}}(j,k;\mathcal{B}_t) - \pi_k f^*(A_{jk}^{t-1})\hat{\mathbb{P}}_{\hat{\mathbf{y}}_{\boldsymbol{\theta}}}(j;\mathcal{B}_t)\right)$
5: $\quad A_{jk}^t = A_{jk}^{t-1} + \eta_{\alpha} \nabla_{\mathbf{A}}\left(A_{jk}^{t-1}\hat{\mathbb{P}}_{\hat{\mathbf{y}}_{\boldsymbol{\theta}},\mathbf{s}}(j,k;\mathcal{B}_t) - \pi_k f^*(A_{jk}^{t-1})\hat{\mathbb{P}}_{\hat{\mathbf{y}}_{\boldsymbol{\theta}}}(j;\mathcal{B}_t)\right)$
6: **Return:** $\boldsymbol{\theta}^T$

---

**Theorem 2.2.** *(**Informal Statement**) Assume that $\ell(\cdot, \cdot)$ and $\mathcal{F}_j(\cdot, \boldsymbol{\theta})$ are Lipschitz continuous for any given $j$ and $\boldsymbol{\theta}$ and their gradients are L-Lipschitz. Further, assume that $\mathbb{P}(s=k) > 0$ for all protected groups and $\mathbb{P}(\hat{y}_{\boldsymbol{\theta}} = j) > 0$ at every iteration for all labels $j$. Then, for any given batch size $1 \leq \lvert\mathcal{B}\rvert \leq n$, Algorithm 1 finds an $\epsilon$-stationary solution of ($f$-FERM) in $\mathcal{O}(\frac{1}{\epsilon^8})$ for any given $\epsilon > 0$.*

*Proof.* The formal statement and proof are relegated to Appendix F. $\qquad\square$

Theorem 2.2 applies to all $f$-divergences listed in Table 1 for all batch-sizes (even as small as the batch size of 1). More sophisticated algorithms can be used to obtain $\mathcal{O}(\epsilon^{-6})$ iteration complexity Rafique et al. (1810); Zhang et al. (2022). However, such algorithms use nested loops and require more hyperparameter tunings. We provide an example of such an algorithm in Appendix G. If the f-divergence leads to a strongly concave function in $\mathbf{A}$ or satisfies Polyak-Łojasiewicz condition (e.g., for $\chi^2$ divergence), a faster rate of $\mathcal{O}(\epsilon^{-5})$ can be obtained for this algorithm (Appendix F). In addition, if larger batch size of $\mathcal{O}(\epsilon^{-2})$ is used, we can further improve this rate to $O(\epsilon^{-4})$ iteration complexity (see Appendix F). Finally, when full batch size is used, then double/triple-loop algorithms can lead to the iteration complexity bounds of $O(\epsilon^{-2})$ in the nonconvex-strongly concave setting and $O(\epsilon^{-3})$ in the general nonconvex-concave setting; see (Kong & Monteiro, 2021; Nouiehed et al., 2019; Ostrovskii et al., 2021b; Thekumparampil et al., 2019).

## 3 ROBUST $f$-FERM IN THE PRESENCE OF DISTRIBUTION SHIFTS

In the previous section, we assumed that the training and test domains have the same distribution. However, this assumption is not necessarily valid in certain applications (Fang et al., 2020). In

particular, a model that behaves fairly on the training data distribution may have an unfair performance in the test phase. To address this issue, this section develops stochastic algorithms for fair empirical risk minimization via $f$-divergences in the presence of the distribution shifts.

Assume that $\hat{\mathbb{P}}_{s,y}(s, \hat{y})$ is the joint distribution of sensitive attributes and predictions on the training data. The distributionally robust fair empirical risk minimization via $f$-divergences is formulated as:

$$\min_{\boldsymbol{\theta}} \frac{1}{n} \sum_{i=1}^{n} \ell(\hat{y}_{\boldsymbol{\theta}}(\mathbf{x}_i), y_i) \quad \text{s.t.} \quad \max_{\mathbb{P} \in \mathcal{B}} \mathcal{D}_f \Big( \mathbb{P}(\hat{y}_{\boldsymbol{\theta}}(\mathbf{x}), s) || \mathbb{P}(\hat{y}_{\boldsymbol{\theta}}(\mathbf{x})) \otimes \mathbb{P}(s) \Big) \leq \kappa. \quad (6)$$

$\mathcal{B} = \mathcal{B}(\hat{\mathbb{P}}, \delta)$ is the distributional uncertainty set defined as a certain ball around the training distribution $\hat{\mathbb{P}}$ with radius $\delta$. This formulation guarantees that the model fairness is preserved (up to a violence of f-divergence less than $\kappa$) even when the test distribution slightly changes. With a slight change of notation, $\hat{\mathbb{P}}$ refers to the training distribution, whereas $\mathbb{P}$ is the optimization parameter.

One can define the uncertainty set through an $\epsilon$ neighborhood around the joint distribution of the training data characterized by a distance measure such as $\ell_p$ norms, Wasserstein distance, or MMD distance. While these distributionally robust uncertainty sets are thoroughly analyzed for empirical risk minimization (ERM) (Kuhn et al., 2019; Blanchet et al., 2019; Levy et al., 2020), the DRO formulation for ERM is limited to the Wasserstein distance for the fair logistic regression (Taskesen et al., 2020) and MMD distance (Wang et al., 2023) on the distribution curvature as a heuristic for robustness. Unfortunately, none of these approaches offer a convergent algorithm with stochastic updates. Further, some of these approaches are limited to special loss functions and heuristics. On the other hand, we study imposing the distributionally robust fairness via $f$-divergences for a general loss function where the uncertainty set is characterized by $\ell_p$ norms (Section 3.1) or $f$-divergences (Section 3.2). Our results show that the former approach is more suitable when lower levels of robustness for fairness are required, and the latter works better for handling larger distribution shifts.

### 3.1 ROBUST $f$-FERM UNDER $\ell_p$ NORMS AND SMALL DISTRIBUTION SHIFTS

This section focuses on the widely studied $\ell_p$ norms as the uncertainty set for the distributional distance between the training and test domains. In this case, Problem (6) can be written as:

$$\min_{\boldsymbol{\theta}} \frac{1}{n} \sum_{i=1}^{n} \ell(\hat{y}_{\boldsymbol{\theta}}(\mathbf{x}_i), y_i) \quad \text{s.t.} \quad \max_{\substack{||\mathbb{P}-\hat{\mathbb{P}}||_p \leq \delta \\ ||\mathbb{Q}-\hat{\mathbb{Q}}||_p \leq \delta}} \mathcal{D}_f(\mathbb{P}||\mathbb{Q}) \leq \kappa, \quad (7)$$

where $\hat{\mathbb{P}}$ represents the joint distribution of the sensitive attributes and predictions and $\hat{\mathbb{Q}}$ denotes the Kronecker product of the marginal distributions between sensitive attributes and predictions.

Since handling non-convex constraints is challenging, as it is standard in training machine learning models, we consider the Lagrangian relaxation of Problem (7) as follows:

$$\min_{\boldsymbol{\theta}} \frac{1}{n} \sum_{i=1}^{n} \ell(\hat{y}_{\boldsymbol{\theta}}(\mathbf{x}_i), y_i) + \lambda \max_{\substack{||\mathbb{P}-\hat{\mathbb{P}}||_p \leq \delta \\ ||\mathbb{Q}-\hat{\mathbb{Q}}||_p \leq \delta}} \mathcal{D}_f(\mathbb{P}||\mathbb{Q}) \quad (8)$$

This problem falls under the nonconvex-nonconcave, min-max optimization category and is most likely to be computationally hard for general uncertainty sets (Daskalakis et al., 2021). However, such a min-max optimization problem can be solved to stationarity when the diameter of set $\mathcal{B}$ is small (i.e., under small domain shift), see (Ostrovskii et al., 2021a). The core idea is to approximate the inner maximization problem with the Taylor approximation, leading to a nonconvex-concave min-max optimization, which is easier to solve (Daskalakis et al., 2021; Razaviyayn et al., 2020b). This idea has been used and been successful in machine learning (see Foret et al. (2020) for its use in Sharpness-aware minimization). Utilizing this idea, Problem (8) can be approximated as:

$$\min_{\boldsymbol{\theta}} \max_{\substack{||\mathbb{U}||_p \leq \delta \\ ||\mathbb{V}||_p \leq \delta}} \left( h(\boldsymbol{\theta}, \mathbb{U}, \mathbb{V}) := \frac{1}{n} \sum_{i=1}^{n} \ell(\hat{y}_{\boldsymbol{\theta}}(\mathbf{x}_i), y_i) + \lambda \langle \mathbb{U}, \nabla_{\mathbb{P}} \mathcal{D}_f(\hat{\mathbb{P}}||\hat{\mathbb{Q}}) \rangle + \lambda \langle \mathbb{V}, \nabla_{\mathbb{Q}} \mathcal{D}_f(\hat{\mathbb{P}}||\hat{\mathbb{Q}}) \rangle \right), \quad (9)$$

where we used the change of variables $\mathbb{U} := \mathbb{P} - \hat{\mathbb{P}}$ and $\mathbb{V} := \mathbb{Q} - \hat{\mathbb{Q}}$. Equivalently,

$$\min_{\boldsymbol{\theta}} \frac{1}{n} \sum_{i=1}^{n} \ell(\hat{y}_{\boldsymbol{\theta}}(\mathbf{x}_i), y_i) + \lambda \delta \|\nabla_{\mathbb{P}} \mathcal{D}_f(\hat{\mathbb{P}}||\hat{\mathbb{Q}})\|_q + \lambda \delta \|\nabla_{\mathbb{Q}} \mathcal{D}_f(\hat{\mathbb{P}}||\hat{\mathbb{Q}})\|_q, \quad (10)$$

where $\| \cdot \|_q$ is the dual of the $\ell_p$ norm with $\frac{1}{p} + \frac{1}{q} = 1$.

**Proposition 3.1.** *Assume that the gradient of the loss function is L-Lipshitz, and the second-order derivative of the loss exists. Then, a given $\epsilon-$approximate stationary solution of Problem* (10) *is an $O(\epsilon)-$approximate stationary solution of Problem* (8) *whenever $L\delta \lesssim \epsilon$.*

This proposition, which is an immediate application of Ostrovskii et al. (2021a, Theorem 3.1), states that if the desired training accuracy $\epsilon$ is comparable with the distribution shift amount $\delta$ (i.e. small distribution shift regime), then one can solve problem (10) instead of (8). Thus, in this regime, we need to solve (10) or equivalently (9). To this end, we need to obtain the (sub)-gradients of the objective function in (9) w.r.t the $\boldsymbol{\theta}$, $\mathbb{U}$, and $\mathbb{V}$ variables. First, notice that

$$\nabla_{\mathbb{U}} h(\boldsymbol{\theta}, \mathbb{U}, \mathbb{V}) = \nabla_{\mathbb{P}} D_f(\hat{\mathbb{P}}||\hat{\mathbb{Q}}) = \boldsymbol{\alpha}^*(\hat{\mathbb{P}}, \hat{\mathbb{Q}}) \text{ and } \nabla_{\mathbb{V}} h(\boldsymbol{\theta}, \mathbb{U}, \mathbb{V}) = \nabla_{\mathbb{Q}} D_f(\hat{\mathbb{P}}||\hat{\mathbb{Q}}) = f^*(\boldsymbol{\alpha}^*(\hat{\mathbb{P}}, \hat{\mathbb{Q}})),$$

where $\boldsymbol{\alpha}^*(\hat{\mathbb{P}}, \hat{\mathbb{Q}}) \in \arg\max_{\boldsymbol{\alpha}} \sum_j \alpha_j \hat{p}_j(\boldsymbol{\theta}) - \hat{q}_j(\boldsymbol{\theta}) f^*(\alpha_j)$. Here we invoked Danskin's theorem on the variational form of $D_f$; $\hat{p}_j(\boldsymbol{\theta})$ and $\hat{q}_j(\boldsymbol{\theta})$ is the $j$-th element of $\hat{\mathbb{P}}$ and $\hat{\mathbb{Q}}$, respectively. Next, we need to compute $\nabla_{\boldsymbol{\theta}} h(\boldsymbol{\theta}, \mathbb{U}, \mathbb{V})$. Notice that the derivative of the first term in $h(\cdot)$ w.r.t. $\boldsymbol{\theta}$ is easy to compute. We next calculate the derivative of the second term of $h(\boldsymbol{\theta}, \mathbb{U}, \mathbb{V})$ w.r.t. $\boldsymbol{\theta}$. As the derivative of the third term can be computed similarly, we omit its derivation here.

$$\nabla_{\boldsymbol{\theta}} \langle \mathbb{U}, \nabla_{\mathbb{P}} \mathcal{D}_f(\hat{\mathbb{P}}||\hat{\mathbb{Q}}) \rangle = \nabla_{\boldsymbol{\theta}} \langle \mathbb{U}, \boldsymbol{\alpha}^*(\hat{\mathbb{P}}, \hat{\mathbb{Q}}) \rangle = \sum_j u_j \frac{\hat{q}_j(\boldsymbol{\theta}) \nabla_{\boldsymbol{\theta}} \hat{p}_j(\boldsymbol{\theta}) - \hat{p}_j(\boldsymbol{\theta}) \nabla_{\boldsymbol{\theta}} \hat{q}_j(\boldsymbol{\theta})}{\hat{q}_j^2(\boldsymbol{\theta}) \times (f^*)''(\alpha)|_{\alpha = \alpha_j^*(\hat{\mathbb{P}}, \hat{\mathbb{Q}})}} \tag{11}$$

where in the last equation, we used the implicit function theorem to compute the derivative of $\boldsymbol{\alpha}^*$ w.r.t. $\boldsymbol{\theta}$. Notice that an implicit assumption here is that $f$ is differentiable (which holds for KL-divergence, $\chi^2$ divergence, reverse KL, Jensen-Shannon, and Squared Hellinger distance). Having access to the gradients, we can apply the standard [sub-]gradient descent-ascent algorithm to obtain a solution to Problem (10) (see Appendix H for the details).

**A semi-stochastic memory-efficient first-order training algorithm.** To apply (stochastic) gradient descent-ascent algorithm (Lin et al., 2020) to problem (9), we need to have unbiased estimator of the function $h(\boldsymbol{\theta}, \mathbb{U}, \mathbb{V})$ w.r.t. $\boldsymbol{\theta}$, $\mathbb{U}$, and $\mathbb{V}$ variables. While it seems challenging to obtain unbiased estimator w.r.t. all variables, one can notice that if $\hat{p}_j(\boldsymbol{\theta})$ and $\hat{q}_j(\boldsymbol{\theta})$ can be computed easily with one forward pass over all data points (i.e., in $O(m \times n)$ memory requirement). Consequently, the gradient of $h(\boldsymbol{\theta}, \mathbb{U}, \mathbb{V})$ w.r.t. $\mathbb{U}$ and $\mathbb{V}$ can be computed with one forward pass over all data points (without the need for doing backpropagation). On the other hand, one can easily obtain unbiased estimator of $\nabla_{\boldsymbol{\theta}} \hat{p}_j(\boldsymbol{\theta})$ and $\nabla_{\boldsymbol{\theta}} \hat{q}_j(\boldsymbol{\theta})$ in (11) using a small mini-batch of data. Such a task requires $O(b \times d)$ memory with $d$ being the number of parameters (i.e., $\boldsymbol{\theta} \in \mathbb{R}^d$) and $b$ being the batch size. Combining this unbiased estimation with the computed values of $\hat{p}_j(\boldsymbol{\theta})$ and $\hat{q}_j(\boldsymbol{\theta})$ leads to an unbiased estimator of the objective of (9) w.r.t. $\boldsymbol{\theta}$ variable. To summarize, we need to do one forward propagation to obtain gradients w.r.t. $\mathbb{U}$ and $\mathbb{V}$, and we only do backpropagation for computing gradients w.r.t. $\boldsymbol{\theta}$ over the mini-batch of data. Such an algorithm requires $O(mn + bd)$ memory requirement and thus can be used for training large models (with $d, n \gg b, m$). It is known that memory requirements are the major limiting factors in training large models such as LLMs (Malladi et al., 2023).

## 3.2 Robust $f$-FERM Under $\ell_\infty$ Norms and Potentially Large Distribution Shifts

The developed framework in the previous section assumes the distribution shift is small (the uncertainty set diameter is smaller than a certain threshold). When preserving fairness in the presence of large distribution shifts is a priority, our previous methodology might not work well. As discussed before, the formulation (8) leads to a nonconvex-nonconcave min-max optimization problem and this class of problems is hard to solve computationally in general (even to stationarity notions). Thus, we need to exploit the structure of the problem. In this section, we show that we can exploit the structure to develop a first-order algorithm under large distribution shifts. Particularly, we focus on the case where the uncertainty set is $\ell_\infty$ ball and the divergence satisfies certain assumptions (i.e., $f^*(\alpha^*) > 0$ and $\alpha^* > 0$, which is satisfied for KL divergence).

Since the function $D_f$ is convex in $\mathbb{P}$ and $\mathbb{Q}$, under $\ell_\infty$ uncertainty set on $\mathbb{P}$ and $\mathbb{Q}$, the optimal solution of the maximization problem in (8) will be at an extreme point. Moreover, under the assumption that $f^*(\alpha^*) > 0$ and $\alpha^* > 0$ (which is satisfied for KL divergence), one can easily see that the optimal $p_j = \min\{\hat{p}_j + \delta, 1\}$ and $q_j = \max\{\hat{q}_j - \delta, 0\}$ (see Appendix I for the exact proof). Notice that we need to relax the probability simplex constraint to obtain this efficient, optimal closed-form solution. Thus under this assumption, problem (8) can be reformulated as

$$\min_{\boldsymbol{\theta}} \frac{1}{n} \sum_{i=1}^n \ell(\hat{y}_{\boldsymbol{\theta}}(\mathbf{x}_i), y_i) + \lambda \mathcal{D}_f(\min\{\mathbb{P} + \delta, 1\} || \max\{\mathbb{Q} - \delta, 0\}), \tag{12}$$

which is a regular minimization problem and (sub)gradient descent can be utilized to solve it.

## 4 EXPERIMENTS

We use three popular notions of group fairness: demographic parity, equalized odds, and equality of opportunity violations (see Appendix A for definitions) to measure the fairness of trained models. To run Algorithm 1, we set $\eta_{\boldsymbol{\theta}}$ and $\eta_{\alpha}$ to $10^{-5}$ and $10^{-6}$ respectively in all experiments. Further, by changing $\lambda$, we get different points in the trade-off curve between accuracy and fairness. The range of $\lambda$ depends on the $f$-divergence (see Appendix J for more information on tuning hyper-parameters). In the inference phase of our experiments, we use the standard maximum likelihood decoding based on the output of the softmax layer, i.e., the predicted label is the label with the highest logit value.

As we will see in this section, several $f$-divergence measures lead to reasonable fairness/accuracy tradeoffs and can outperform existing benchmarks. However, no single $f$-divergence measure uniformly outperforms other measures in all the experiments. Thus, we believe in applications, the choice of the $f$-divergence can be viewed as a hyperparameter that can be tuned by cross-validation.

### 4.1 FAIRNESS-ACCURACY TRADEOFFS ON BENCHMARK DATASETS

In the first set of experiments, we compare different $f$-divergence formulations for ($f$-FERM) to each other and several state-of-the-art approaches supporting multiple sensitive attributes. Figure 1 demonstrates the given tradeoff on the adult dataset (Becker & Kohavi, 1996) with gender and race as the sensitive attributes (black-female, black-male, white-female, white-male). To measure fairness, we use the demographic parity violation defined as:

$$\text{DPV} = \max_{i,j \in \mathcal{S}} |\mathbb{P}(\hat{y} = 1|s = i) - \mathbb{P}(\hat{y} = 1|s = j)|$$

In the case of binary sensitive attributes (e.g., gender), there is no significant variation between different $f$-divergences. However, when we have 2 sensitive attributes and the batch size is small (8 in Figure 1), the results significantly differ for various $f$-divergences. Interestingly, KL-divergence for smaller $\lambda$ values shows improvement in fairness violation and accuracy simultaneously. We do not observe such a phenomenon for other $f$-divergences and state-of-the-art approaches in the literature. Further, in Figure 2, we compare one of the $f$-divergences (reverse KL)

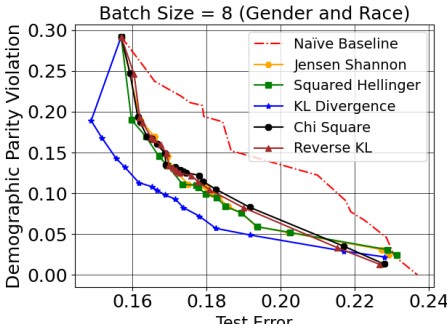

Figure 1: Performance of different $f$-divergences as the regularizers. The experiment is on the adult dataset with gender and race as sensitive attributes. While the offered tradeoffs are close to each other for small demographic parity violations, KL-divergence shows an extraordinary performance for a low-fairness high-accuracy regime. We do not display the performance for larger batch sizes or when only one sensitive attribute is available due to the insignificant difference between the performance of different $f$-divergences.

to several SOTA methods including Mary et al. (2019); Baharlouei et al. (2020); Cho et al. (2020). Other approaches such as the pre-processing method of Zemel et al. (2013), post-processing approach of Hardt et al. (2016), and several in-processing methods including Zafar et al. (2017); Donini et al. (2018); Jiang et al. (2020) demonstrate lower performance compared to the ones depicted in Figure 2 and are removed from the figure. While our approach demonstrates consistently good performance across different batch sizes (full-batch, 64, 8, 2), the performances of other methods drop significantly for smaller ones. For further experiments on other datasets (German and COMPAS) and other fairness measures (equality of opportunity and equalized odds violations), see Appendix K.

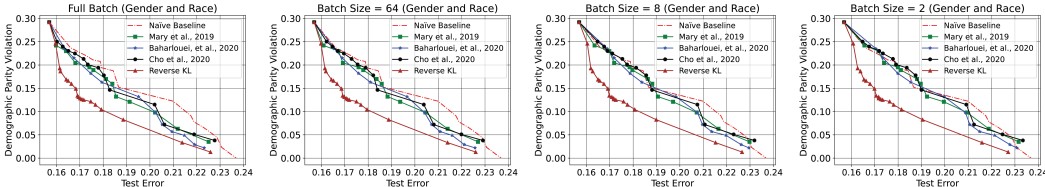

Figure 2: Performance of the trained fair models on Adult Dataset with gender and race as two sensitive attributes with different Batch-sizes. The red dashed line represents the Naïve baseline where the model outputs zero with probability $p$. By increasing $p$, the model becomes fairer at the cost of the loss in accuracy.

### 4.2 FAIRNESS-ACCURACY TRADEOFFS IN THE PRESENCE OF THE DISTRIBUTION SHIFT

We perform two experiments to evaluate the Algorithms developed in Section 3. In the first experiment, we randomly switch the label of genders for $n\%$ of the data points ($n$ ranges from 1 to 20) in the

Adult dataset. Then, we train models on the new datasets with a proportion of corrupted sensitive attributes and evaluate the performance on the test data. Figure 3 is obtained by training different models to achieve $80\%$ accuracy on the test data and comparing their demographic parity violation. By increasing the percentage of corrupted sensitive attributes, we see that both $f$-DRO and $f$-infinity achieve less DP violation than SOTA approaches in the literature. In this specific experiment, $f$-DRO works better than $f$-infinity, and there is no significant difference between choosing KL-divergence or $\chi^2$ as the function $f$. Among the papers designed for handling distribution shifts, Rezaei et al. (2021) and Wang et al. (2020) were the only options with the available implementation.

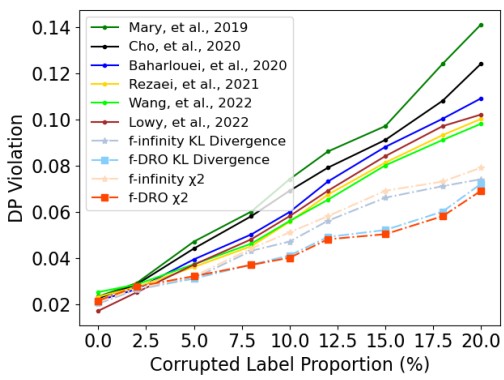

In a more recently collected dataset (new adult) (Ding et al., 2021), the users are separated based on their living state. We train different fair models in a single state and evaluate the fairness-accuracy tradeoff in other states. Figure 4 depicts the performance of different methods. For each method, the center point is the average of accuracy and fairness among 50 states. The horizontal and vertical lines show the 25-percentile to 75-percentile range of performance among the states. The training fairness violation is set to $0.02$ for all methods. We observe that $f$-infinity preserves the fairness level better than other approaches. In comparison, $f$-DRO has a better accuracy. Depending on the application, we suggest using $f$-infinity if preserving a high level of fairness is a priority and $f$-DRO for the cases when a better tradeoff between fairness and accuracy is expected. Note that both approaches offer better fairness-accuracy trade-offs compared to the SOTA approaches in the literature.

Figure 3: Performance of different state-of-the-art approaches and our two methods for handling distribution shift. The dataset is adult, and the sensitive attribute is gender. We randomly flip the label of a proportion of gender entries (from 0 to $20\%$). As we observe, our approach demonstrates more robustness against the drop in DP violation compared to other approaches.

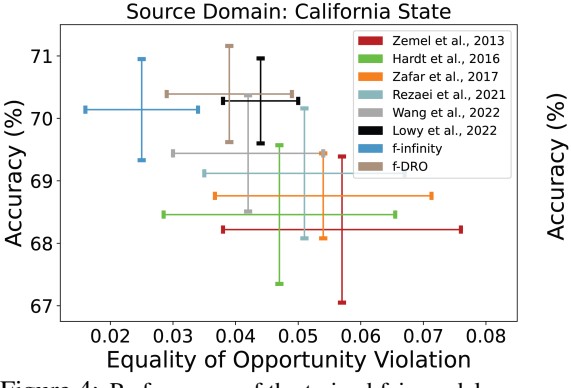
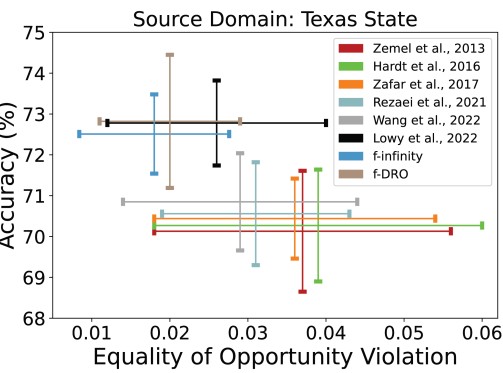

Figure 4: Performance of the trained fair models on new Adult Dataset. The model is trained on one state (California or Texas) and evaluated in 50 states. The distribution of each state dataset is different than others. Thus, the IID assumption does not hold among datasets of different states.

## 5 CONCLUSION

This paper presented a unified stochastic framework for fair empirical risk minimization via $f$-divergences ($f$-FERM). The key idea is to reformulate the objective function as a min-max optimization problem using Legendre-Fenchel duality of $f$-divergence. This enables us to develop an unbiased gradient estimator and a convergent stochastic first-order algorithm. Furthermore, we robustified $f$-FERM using $\ell_p$ norm balls as the uncertainty set against distributional changes. While our empirical investigation delves into the performance and fairness distinctions among various $f$-divergences, a more comprehensive analysis is warranted to determine the optimal $f$-divergence concerning the tradeoff between performance and fairness, faster convergence, and asymptotic behaviors. Furthermore, the distributionally robust formulation of fair empirical risk minimization and the advantages of each formulation can be explored beyond $f$-divergences as the measure of fairness violation and $\ell_p$ norm balls as uncertainty sets.

ACKNOWLEDGEMENTS

This work was supported by the NSF CAREER Award CCF2144985, the AFOSR Young Investigator Program Award FA9550-22-1-0192, a gift from the USC-Meta Center for Research and Education in AI, and a gift from Google.

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
