## A $f$-FERM FOR OTHER NOTIONS OF GROUP FAIRNESS

This section shows how we can use alternative notions of fairness, such as equality of opportunity of equalized odds (Hardt et al., 2016) instead of demographic parity violation in $f$-FERM.

Note that a trained model satisfies the equality of opportunity notion for a given binary classifier with a binary sensitive attribute if and only if:

$$\mathbb{P}(\hat{y}_{\boldsymbol{\theta}}(\mathbf{x}) = 1, s = i|\, y = 1) = \mathbb{P}(\hat{y}_{\boldsymbol{\theta}}(\mathbf{x}) = 1, s = j|\, y = 1) \quad \forall\, i, j \in \mathcal{S} \tag{13}$$

Therefore, to have a framework for fair inference via $f$-divergences under the equality of opportunity notion, we optimize:

$$\min_{\boldsymbol{\theta}} \quad \frac{1}{n} \sum_{i=1}^{n} \ell(\hat{y}_{\boldsymbol{\theta}}(\mathbf{x}_i), y_i) + \lambda \mathcal{D}_f \Big( \mathbb{P}(\hat{y}_{\boldsymbol{\theta}}(\mathbf{x}), s|y = 1) || \mathbb{P}(\hat{y}_{\boldsymbol{\theta}}(\mathbf{x})|y = 1) \otimes \mathbb{P}(s|y = 1) \Big). \tag{14}$$

Practically, it means that for evaluating the probability measures in the regularization term, we need to focus on the data points whose target labels are 1.

Further, one can similarly adopt equalized odds as the measure of fairness. Equalized odds as the measure of fairness is defined as:

$$\mathbb{P}(\hat{y}_{\boldsymbol{\theta}}(\mathbf{x}) = 1, s = i|\, y = k) = \mathbb{P}(\hat{y}_{\boldsymbol{\theta}}(\mathbf{x}) = 1, s = j|\, y = k) \quad \forall\, i, j \in \mathcal{S}, \ k \in \mathcal{Y} \tag{15}$$

Therefore, we must add a regularizer per each class label to satisfy the equalized odds notion. Other notions of fairness can be used in this framework as long as they can be represented as the conditional independence between sensitive attributes, predictions, and labels (Castelnovo et al., 2022).

## B $f$-DIVERGENCES FOR CONTINUOUS SENSITIVE ATTRIBUTES AND TARGET VARIABLES

In Section 2, we developed a framework for promoting fairness for classification problems where both target labels and sensitive attributes are discrete variables. Hence, we could efficiently solve the variational formulation that arose through the designing of unbiased estimators. However, it is not uncommon to find applications of f-divergence regularizers in practice that require either the sensitive features or the output variable to be continuous or both to be continuous parameters. In such cases, the summation over the respective variable is replaced by an integral over the probability distribution. The challenging aspect is calculating the variational form's integral and trailing supremum in the continuous domain.

Let $P$ and $Q$ be two continuous distributions over the space $\Omega$ such that $P$ is absolutely continuous with respect to $Q$ ($P \ll Q$). Then, the $f$-divergence between these two distributions for a given convex function $f$ is defined as:

$$\mathcal{D}_f(\mathbb{P}, \mathbb{Q}) = \int_{\Omega} f\Big(\frac{dP}{dQ}\Big) dQ \tag{16}$$

When the target variable is continuous (regression problems), but the sensitive attribute is discrete, ($f$-FERM) can be written as:

$$\min_{\boldsymbol{\theta}} \max_{\mathbf{A} \in \mathbb{R}^{\infty}} \sum_{i=1}^{n} \ell(\hat{y}_{\boldsymbol{\theta}}(\mathbf{x}_i), y_i) + \lambda \sum_k \int_x \Big[ \mathbf{A}_k(x)\mathbb{P}(x) - f^*(\mathbf{A}_k(x))\mathbb{Q}_k \Big] dx$$

With slight changes, the above problem can be reformulated as follows:

$$\min_{\boldsymbol{\theta}} \max_{\mathbf{A} \in \mathbb{R}^{jk}} \sum_{i=1}^{n} \ell(\hat{y}_{\boldsymbol{\theta}}(\mathbf{x}_i), y_i) + \lambda \max_{\mathbf{A}_1, \dots, \mathbf{A}_m} \sum_k \mathbb{E}\Big[ \mathbf{A}_k(s)\mathbb{P}_j(s) - f^*(\mathbf{A}_k(s))\mathbb{Q}_k \Big]$$

When both sensitive features and target variables are continuous, the objective function becomes:

$$\min_{\boldsymbol{\theta}} \max_{\mathbf{A} \in \mathbb{R}^{\infty \times \infty}} \sum_{i=1}^{n} \ell(\hat{y}_{\boldsymbol{\theta}}(\mathbf{x}_i), y_i) + \lambda \int_x \int_y \Big[ \mathbf{A}(x, y)\mathbb{P}(x) - f^*(\mathbf{A}(x, y))\mathbb{Q}(y) \Big] dx\, dy$$

Such a formulation is clearly intractable for solving $\mathbf{A}_k(x)$ or $\mathbf{A}(x, y)$ in the continuous domain. We need to approximate the above integrals in discretized/quantized regions or find another variational representation for designing unbiased estimators of continuous domain $f$-divergences. We leave developing algorithms for the continuous target variables and sensitive attributes as a future direction.

## C $f$-DIVERGENCES COVER WELL-KNOWN NOTIONS OF FAIRNESS VIOLATION

In this section, first, we show that optimizing $f$-divergences to $0$ guarantees the independence of the sensitive attributes and predictions. In other words, optimizing $f$-divergences leads to a fair model under the demographic parity notion (or other group fairness notions discussed in Appendix A).

**Proposition C.1.** *(Polyanskiy & Wu, 2022, Theorem 2.3) Let $f$ be a convex function from $\mathbb{R}^+$ to $\mathbb{R}$, such that $f$ is convex, $f(1) = 0$, $f$ is strictly convex in a neighborhood of $1$. Then $\mathcal{D}_f(\mathbb{P}||\mathbb{Q}) = 0$, if and only if $P = Q$.*

As an immediate result, a trained model in ($f$-FERM) is fair under the demographic notion if and only if

$$\mathbb{P}(\hat{y}_{\boldsymbol{\theta}}(\mathbf{x}), s) = \mathbb{P}(\hat{y}_{\boldsymbol{\theta}}(\mathbf{x})) \otimes \mathbb{P}(s), \tag{17}$$

which means the independence of $s$ and $\hat{y}_{\boldsymbol{\theta}}(\mathbf{x})$.

Next, we show $f$-divergences either include or provide an upper bound for well-known notions of fairness violation in the literature.

**Proposition C.2.** *Exponential Rényi Mutual Information (ERMI) (Mary et al., 2019; Lowy et al., 2022) is an $f$-divergence with $f(t) = (t-1)^2$*

*Proof.* Exponential Rényi Mutual Information is defined as (Lowy et al., 2022):

$$\mathrm{ERMI}(\hat{y}, s) = \sum_{j \in \mathcal{Y}, k \in \mathcal{S}} \frac{\hat{P}_{\hat{y},s}(j,k)^2}{\hat{P}_{\hat{y}}(j)\hat{P}_s(k)} - 1 \tag{18}$$

For the case of $f(t) = (t-1)^2$, we have:

$$\mathcal{D}_f\left(\hat{P}_{\hat{y}} \otimes \hat{P}_s || \hat{P}_{\hat{y},s}\right) = \sum_{j \in \mathcal{Y}} \sum_{k \in \mathcal{S}} \hat{P}_{\hat{y}}(j)\hat{P}_s(k) f\left(\frac{\hat{P}_{\hat{y},s}(j,k)}{\hat{P}_{\hat{y}}(j)\hat{P}_s(k)}\right) = \sum_{j \in \mathcal{Y}} \sum_{k \in \mathcal{S}} \hat{P}_{\hat{y}}(j)\hat{P}_s(k) \left(\frac{\hat{P}_{\hat{y},s}(j,k)}{\hat{P}_{\hat{y}}(j)\hat{P}_s(k)} - 1\right)^2$$

$$= \sum_{j \in \mathcal{Y}} \sum_{k \in \mathcal{S}} \hat{P}_{\hat{y}}(j)\hat{P}_s(k) \left(\frac{\hat{P}_{\hat{y},s}(j,k)^2}{\hat{P}_{\hat{y}}(j)^2 \hat{P}_s(k)^2} - 2\frac{\hat{P}_{\hat{y},s}(j,k)}{\hat{P}_{\hat{y}}(j)\hat{P}_s(k)} + 1\right)$$

$$= \sum_{j \in \mathcal{Y}} \sum_{k \in \mathcal{S}} \left(\frac{\hat{P}_{\hat{y},s}(j,k)^2}{\hat{P}_{\hat{y}}(j)\hat{P}_s(k)} - 2\hat{P}_{\hat{y},s}(j,k) + \hat{P}_{\hat{y}}(j)\hat{P}_s(k)\right)$$

$$= \sum_{j \in \mathcal{Y}} \sum_{k \in \mathcal{S}} \frac{\hat{P}_{\hat{y},s}(j,k)^2}{\hat{P}_{\hat{y}}(j)\hat{P}_s(k)} - 2 + 1 = \mathrm{ERMI}(\hat{y}, s)$$

Note that, in the last equality, we use:

$$\sum_{j \in \mathcal{Y}} \sum_{k \in \mathcal{S}} \hat{P}_{\hat{y},s}(j,k) = \sum_{j \in \mathcal{Y}} \hat{P}_{\hat{y}}(j) = 1,$$

and

$$\sum_{j \in \mathcal{Y}} \sum_{k \in \mathcal{S}} \hat{P}_{\hat{y}}(j)\hat{P}_s(k) = \sum_{j \in \mathcal{Y}} \hat{P}_{\hat{y}}(j)\left(\sum_{k \in \mathcal{S}} \hat{P}_s(k)\right) = \sum_{j \in \mathcal{Y}} \hat{P}_{\hat{y}}(j) = 1,$$

$\square$

**Proposition C.3.** *Demographic parity violation is upper-bounded by the $f$-divergence for $f(t) = (t-1)^2$*

*Proof.* Based on Propositions C.2, ERMI is an $f$-divergence with $f(t) = (t-1)^2$. Therefore, the proposition is an immediate result of Lemma 3 in (Lowy et al., 2022). $\square$

**Proposition C.4.** *Rényi correlation (Baharlouei et al., 2020) can be upper bounded by the $f$-divergences for the choice of $f(t) = (t-1)^2$.*

*Proof.* Based on Propositions C.2, ERMI is an $f$-divergence with $f(t) = (t-1)^2$. Therefore, the proposition is an immediate result of Lemma 2 in (Lowy et al., 2022). $\square$

*Remark* C.5. Mutual Information as the measure of fairness violation (Cho et al., 2020) is a special case of $f$-divergences for the choice of KL-divergence $f(t) = t \log(t)$ in ($f$-FERM).

## D    PROOF OF PROPOSITION 2.1

**Lemma D.1.** *Assume that $f(\mathbf{z})$ is a semi-continuous convex function. Therefore, $f$ can be written as the following maximization problem:*

$$f(\mathbf{z}) = \max_{\alpha} \mathbf{z}^T \alpha - g(\alpha)$$

*where $g$ is the convex conjugate of $f$.*

*Proof.* Let $g$ be the convex conjugate of the function $f$ defined as:

$$g(\alpha) = \sup_{\mathbf{z}} \boldsymbol{\alpha}^T \mathbf{z} - f(\mathbf{z})$$

Since $f$ is a lower semi-continuous convex function, by Fenchel-Moreau theorem (Ioffe & Tihomirov, 2009), it is biconjugate, which means the taking conjugate of $g$ transforms it back to $f$. Therefore,

$$f(\mathbf{z}) = \sup_{\boldsymbol{\alpha}} \boldsymbol{\alpha}^T \mathbf{z} - g(\boldsymbol{\alpha})$$

where $g$ is the convex conjugate of $f$. ∎

Based on the above lemma, we have:

$$\mathcal{D}_f(\mathbb{P},\mathbb{Q}) = \sum_{i=1}^{m} \mathbb{Q}_i f\left(\frac{\mathbb{P}_i}{\mathbb{Q}_i}\right) = \mathcal{D}_f(\mathbb{P},\mathbb{Q}) = \sum_{i=1}^{m} \mathbb{Q}_i \sup_{\alpha_i \in \mathrm{dom} f} \alpha_i \frac{\mathbb{P}_i}{\mathbb{Q}_i} - f^*(\alpha_i)$$

$$= \sup_{\alpha_1,\ldots,\alpha_m \in \mathrm{dom} f} \sum_{i=1}^{m} \alpha_i \mathbb{P}_i - f^*(\alpha_i)\mathbb{Q}_i$$

Set $\mathbb{P} = \mathbb{P}(\hat{y}_{\boldsymbol{\theta}}(\mathbf{x}), s)$, $\mathbb{Q} = \mathbb{P}(\hat{y}_{\boldsymbol{\theta}}(\mathbf{x})) \otimes \mathbb{P}(s)$, and $\alpha_i = A_{jk}$. Therefore, we obtain the formulation in (3).

## E    DERIVATION OF CLOSED-FORM EXPRESSIONS FOR UNBIASED GRADIENT ESTIMATORS OF $f$-DIVERGENCES

**Proposition E.1.** *For two functions $f(t), g(t)$ such that $g(t) = f(t) + c(t-1)$, then $\mathcal{D}_f(\cdot|\cdot) \equiv \mathcal{D}_g(\cdot|\cdot)$.*

*Proof.* Proof follows naturally from (Polyanskiy & Wu, 2022, Proposition 7.2) ∎

**Theorem E.2.** *Let $f(t) = (t-1)^2$ and $\mathbb{P}(s = k) = \pi_k$ ($\chi^2$ Divergence). Then, Equation (1) can be written as:*

$$\min_{\boldsymbol{\theta}} \max_{\mathbf{A}} \sum_{i=1}^{n} \ell(\hat{y}_{\boldsymbol{\theta}}(\mathbf{x}_i), y_i) + \lambda \sum_j \sum_k \pi_k \left[ A_{jk}\mathbb{P}(\hat{y}_{\boldsymbol{\theta}} = j|s = k) - (A_{jk} + \frac{A_{jk}^2}{4})\mathbb{P}(\hat{y}_{\boldsymbol{\theta}} = j) \right] \tag{19}$$

Variational Representation of $f(x) = (x-1)^2$ is given by

$$f(x) = \sup_{\alpha}(\alpha x - f^*(\alpha))$$

Where $f^*(\alpha)$ is the convex conjugate

$$f^*(\alpha) = \sup_x(x\alpha - f(x))$$

Taking derivative of $f^*(\alpha)$ w.r.t x gives $x^* = \alpha/2 + 1$. This results in $f^*(\alpha) = \alpha + \alpha^2/4$

**Theorem E.3.** *Let $f(t) = -\ln(t)$ and $\mathbb{P}(s = k) = \pi_k$ (Reverse KL). Then, Equation (1) can be written as:*

$$\min_{\boldsymbol{\theta}} \max_{\mathbf{A}} \sum_{i=1}^{n} \ell(\hat{y}_{\boldsymbol{\theta}}(\mathbf{x}_i), y_i) + \lambda \sum_j \sum_k \pi_k \left[ A_{jk}\mathbb{P}(\hat{y}_{\boldsymbol{\theta}} = j|s = k) + (1+\ln(-A_{jk}))\mathbb{P}(\hat{y}_{\boldsymbol{\theta}} = j) \right] \tag{20}$$

Proceeding as above, optimal $x^*$ for the supremum of $f^*(\alpha)$ is $x^* = -1/\alpha$, resulting in $f^*(\alpha) = -1 - \ln(-\alpha)$.

**Theorem E.4.** *Let $f(t) = \frac{1}{2}|t - 1|$ and $\mathbb{P}(s = k) = \pi_k$ (Total Variational Distance). Then, Equation* (1) *can be written as (where $|A_{jk}| \leq \frac{1}{2}$):*

$$\min_{\boldsymbol{\theta}} \max_{\mathbf{A}} \sum_{i=1}^{n} \ell(\hat{y}_{\boldsymbol{\theta}}(\mathbf{x}_i), y_i) + \lambda \sum_j \sum_k \pi_k A_{jk} \Big[ \mathbb{P}(\hat{y}_{\boldsymbol{\theta}} = j | s = k) - \mathbb{P}(\hat{y}_{\boldsymbol{\theta}} = j) \Big] \quad (21)$$

For $f = \frac{1}{2}|t - 1|$, the variational representation is $f(x) = \sup_\alpha \left( \alpha x - f^*(\alpha) \right)$

Through the convex conjugate $f^*(\alpha)$, we have that

$$f^*(\alpha) = \sup_x \left( x\alpha - f(\alpha) \right) = \sup_x \left( x\alpha - \frac{1}{2}|x - 1| \right)$$

$$= \begin{cases} \infty & \text{for } |\alpha| > \frac{1}{2} \\ \alpha & \text{for } |\alpha| \leq \frac{1}{2} \end{cases}$$

So $|\alpha| \leq \frac{1}{2}$ is constrained for the supremum/maximum to exist (otherwise tends to $\infty$).

**Theorem E.5.** *Let $f(t) = t\ln(t)$ and $\mathbb{P}(s = k) = \pi_k$ (KL Divergence). Then, Equation* (1) *can be written as:*

$$\min_{\boldsymbol{\theta}} \max_{\mathbf{A}} \sum_{i=1}^{n} \ell(\hat{y}_{\boldsymbol{\theta}}(\mathbf{x}_i), y_i) + \lambda \sum_j \sum_k \pi_k \Big[ A_{jk}\mathbb{P}(\hat{y}_{\boldsymbol{\theta}} = j | s = k) - e^{A_{jk}-1}\mathbb{P}(\hat{y}_{\boldsymbol{\theta}} = j) \Big] \quad (22)$$

For $f(t) = t\ln(t)$ in f-divergence, the convex conjugate can be represented by:

$$f^*(\alpha) = \sup_x (x\alpha - x\ln(x))$$

On differing w.r.t $x$ for attaining supremum, we get $x = e^{\alpha-1}$. Hence, the variational representation of $f(t) = t\ln(t)$ becomes:

$$f(x) = \sup_\alpha \left( x\alpha - e^{\alpha-1} \right)$$

**Note:** We can also use the affine transformation $\alpha \leftarrow \alpha - 1$, which results in the more commonly studied version in literature:

$$D(P||Q) = 1 + \sup_{g:X \to \mathbb{R}} \mathbb{E}_P[g(X)] - \mathbb{E}_Q[e^{g(X)}]$$

**Theorem E.6.** *Let $f(t) = -(t + 1)\ln(\frac{t+1}{2}) + t\ln(t)$ and $\mathbb{P}(s = k) = \pi_k$ (Jensen-Shannon Divergence). Then, Equation* (1) *can be written as:*

$$\min_{\boldsymbol{\theta}} \max_{\mathbf{A}} \sum_{i=1}^{n} \ell(\hat{y}_{\boldsymbol{\theta}}(\mathbf{x}_i), y_i) + \lambda \sum_j \sum_k \pi_k \Big[ A_{jk}\mathbb{P}(\hat{y}_{\boldsymbol{\theta}} = j | s = k) + \ln(2 - e^{A_{jk}})\mathbb{P}(\hat{y}_{\boldsymbol{\theta}} = j) \Big] \quad (23)$$

For the JS Divergence, we have $f(t) = -(t + 1)\ln(\frac{t+1}{2}) + t\ln(t)$, whose convex conjugate can be represented as:

$$f^*(\alpha) = \sup_x \left( \alpha x + (x + 1)\ln\left(\frac{x+1}{2}\right) - x\ln(x) \right)$$

On differentiating w.r.t $x$ to obtain the supremum, we have

$$\frac{2x}{x+1} = e^\alpha \implies x = \frac{e^\alpha}{2 - e^\alpha}$$

Substituting $x$ in $f^*(\alpha)$,

$$f^*(\alpha) = -\ln(2 - e^\alpha)$$

Thus, in $f(x) = \sup_\alpha \left( x\alpha - f^*(\alpha) \right)$, we get the variational form as:

$$f(x) = \sup_\alpha \left( x\alpha + \ln(2 - e^\alpha) \right)$$

**Theorem E.7.** *Let $f(t)$ be*

$$f(t) = \begin{cases} \frac{t^\alpha - \alpha t - (1-\alpha)}{\alpha(\alpha-1)} & \text{if } \alpha \neq 0, \alpha \neq 1 \\ t\ln(t) - t + 1 & \text{if } \alpha = 1 \\ -\ln(t) + t - 1 & \text{if } \alpha = 0 \end{cases}$$

*and $\mathbb{P}(s = k) = \pi_k$ (General $\alpha$ Divergence). Then, Equation (1) can be written as:*

$$\min_{\boldsymbol{\theta}} \max_{\mathbf{A}} \sum_{i=1}^n \ell(\hat{y}_{\boldsymbol{\theta}}(\mathbf{x}_i), y_i) + \lambda \sum_j \sum_k \pi_k \Big[ A_{jk}\mathbb{P}(\hat{y}_{\boldsymbol{\theta}} = j | s = k)$$
$$- \frac{\mathbb{P}(\hat{y}_{\boldsymbol{\theta}} = j)}{\alpha} \Big( \big((\alpha-1)A_{jk} + 1\big)^{\frac{\alpha}{\alpha-1}} - 1 \Big) \Big] \quad (24)$$

Excluding the limiting cases where $\alpha = 1$ or $\alpha = 0$, we can find the convex conjugate $f^*(y)$ as:

$$f^*(y) = \sup_x \Big( xy - f(x) \Big)$$
$$= \sup_x \Big( xy - \frac{x^\alpha - \alpha x - (1-\alpha)}{\alpha(\alpha-1)} \Big)$$

On differentiating w.r.t. $x$, we obtain (here variational parameter is $y$, do not confuse with the constant $\alpha$)

$$x^* = \Big( (\alpha-1)y + 1 \Big)^{\frac{1}{\alpha-1}}$$

Thus,

$$f^*(y) = \frac{\Big( (\alpha-1)y + 1 \Big)^{\frac{\alpha}{\alpha-1}}}{\alpha} - \frac{1}{\alpha}$$

KL Divergence and Reverse KL Divergence can be obtained by taking the limit when $\alpha$ tends to 1 and 0, respectively.

**Note:** Standard literature on divergences often parametrize the $\alpha$-divergence as

$$f(x) = \begin{cases} t\ln(t) & \text{if } \alpha = 1 \\ -\ln(t) & \text{if } \alpha = -1 \\ \frac{4}{1-\alpha^2}\Big( 1 - t^{(1+\alpha/2)} \Big) & \text{otherwise} \end{cases}$$

This is equivalent to the substitution $\alpha \leftarrow \frac{1+\alpha}{2}$ in the original definition of generalized f-divergence.

**Theorem E.8.** *Let $f(t) = (\sqrt{t} - 1)^2$ (equivalently $f(t) = 2(1 - \sqrt{t})$) and $\mathbb{P}(s = k) = \pi_k$ (Squared Hellinger Distance). Then, Equation (1) can be written as:*

$$\min_{\boldsymbol{\theta}} \max_{\mathbf{A}} \sum_{i=1}^n \ell(\hat{y}_{\boldsymbol{\theta}}(\mathbf{x}_i), y_i) + \lambda \sum_j \sum_k \pi_k \Big[ A_{jk}\mathbb{P}(\hat{y}_{\boldsymbol{\theta}} = j | s = k) + \mathbb{P}(\hat{y}_{\boldsymbol{\theta}} = j)\Big( \frac{1}{A_{jk}} + 2 \Big) \Big] \quad (25)$$

For Squared Hellinger Distance,

$$f^*(\alpha) = \sup_x (x\alpha - f(x))$$
$$= \sup_x (x\alpha - 2(1 - \sqrt{x}))$$

On differentiating w.r.t. $x$, we get

$$\alpha + \frac{1}{\sqrt{x}} = 0 \text{ (Note } \alpha < 0) \implies x = \frac{1}{(\alpha)^2}$$
$$\implies f^*(\alpha) = \frac{\alpha}{\alpha^2} - 2 + \frac{(-2)}{\alpha} = \frac{-1}{\alpha} - 2$$

Note that the first, second, and third terms are negative, negative, and positive, respectively; hence, the appropriate choice of $\text{sign}(\alpha)$ for functions of odd powers of $\alpha$.

## F  FORMAL STATEMENT OF THEOREM 2.2 AND PROOF

**Theorem F.1.**  *Formal Statement of Theorem Let* $(\mathbf{x}_i, y_i, s_i)$   $\forall 1 \leq i \leq n$ *be the collection of* $n$ *data points satisfying the following assumptions:*

- $\ell(\cdot, \mathbf{x}, y)$ *is G-Lipschitz, and* $\beta_\ell$-*smooth for all* $\mathbf{x}_i, y_i$.

- $F_j(\cdot, \boldsymbol{\theta})$ *is L-Lipschitz and b-smooth for all* $\boldsymbol{\theta}$ *and all label classes* $j$.

- $\widehat{p}_{\hat{y}}^{\min} := \inf_{\{\boldsymbol{\theta}^t, t \in [T]\}} \min_{j \in [m]} \frac{1}{N} \sum_{i=1}^{N} \hat{y}_{\boldsymbol{\theta}, j}(\mathbf{x}_i) \geq \frac{\mu}{2} > 0.$

- $\hat{p}_S^{\min} := \frac{1}{N} \sum_{i=1}^{N} \mathbb{1}_{\{s_i = j\}} > 0.$

*choose* $\eta_{\boldsymbol{\theta}} = \Theta(\frac{\epsilon^4}{\ell^3 L^2 D^2})$ *and* $\eta_\alpha = \Theta(\frac{\epsilon^2}{\ell \sigma^2})$ *and the mini-batch size of* $1$. *Therefore, Algorithm 1 finds an* $\epsilon$-*stationary of Problem* $f$-*FERM in* $\mathcal{O}(\frac{1}{\epsilon^8})$.

*Remark* F.2.  The **first assumption** listed in the theorem statement is true for popular losses such as cross-entropy loss and squared loss (assuming that the input data takes values in a bounded set, which holds for all real-world datasets).

*Remark* F.3.  The **second assumption** holds for popular classifiers generating probability vectors (e.g., logits in neural networks, logistic regression outputs). For classifiers with no probability output, one must transform the output to a number between zero and one first.

*Remark* F.4.  The **third assumption** states that the probability of assigning a label to the data points must not be zero for all data points for any label at each iteration.

*Remark* F.5.  Finally, the **fourth assumption** ensures each sensitive class's probability is not zero. In other words, there should be at least one point in the dataset with that sensitive attribute for any sensitive group. It holds for all benchmark datasets in practice. Simply put, any protected group appearing during the test phase must have at least one representative in the training data.

The following lemma is helpful for the proof of the theorem:

**Lemma F.6.** *Let* $A_1, \dots, A_n$ *be* $n$ *variables such that* $\|A_i\|_2 \leq c_i$. *Then, we have:*

$$\mathbb{E}[\|\sum_{i=1}^{n} A_i\|_2^2] \leq n \sum_{i=1}^{n} c_i^2 \tag{26}$$

*Proof.*

$$\|\sum_{i=1}^{n} A_i\|_2^2 = \sum \|A_i\|_2^2 + 2 \sum_{i \neq j} \langle A_i, A_j \rangle \leq \sum \|A_i\|_2^2 + \sum_{i \neq j} \|A_i\|_2^2 + \|A_j\|_2^2 = n \sum_{i=1}^{n} \|A_i\|_2^2,$$

which is based on the fact that $2\langle A_i, A_j \rangle \leq \|A_i\|_2^2 + \|A_j\|_2^2$. Therefore:

$$\mathbb{E}[\|\sum_{i=1}^{n} A_i\|_2^2] \leq n \sum_{i=1}^{n} \mathbb{E}[\|A_i\|_2^2] \leq n \sum_{i=1}^{n} c_i^2$$

$\square$

Now, we are ready to prove Theorem F.1.

*Proof.*  The proof consists of three main steps. First, we need to show that the gradient estimator in Algorithm 1 is unbiased. Since the samples are IID, for any function $\psi(\cdot, \cdot)$, and an IID batch of data points $\mathcal{B}$ we have:

$$\mathbb{E}\Big[\frac{1}{\mathcal{B}} \sum_{(\mathbf{x}, y) \in \mathcal{B}} \nabla \psi(\mathbf{x}, y)\Big] = \frac{1}{\mathcal{B}} \sum_{(\mathbf{x}, y)} \mathbb{E}[\psi(\mathbf{x}, y)] = \mathbb{E}_{(\mathbf{x}, y) \sim \mathbb{P}(\mathbf{x}, y, s)}[\nabla \psi(\mathbf{x}, y)]$$

As an immediate result, if the objective function is written as the summation over $n$ functions, the gradient estimator over an IID batch of data will be unbiased. According to Equation (5), the objective function has the desired form for:

$$\min_{\boldsymbol{\theta}} \max_{\mathbf{A}} \quad \frac{1}{n} \sum_{i=1}^{n} \left[ \ell(\hat{y}_{\boldsymbol{\theta}}(\mathbf{x}_i), y_i) + \lambda \sum_{\substack{j \in \mathcal{Y}, \\ k \in \mathcal{S}}} \left[ A_{jk} F_j(\mathbf{x}_i; \boldsymbol{\theta}) \mathbb{1}(s_i = k) - f^*(A_{jk}) \pi_k F_j(\mathbf{x}_i; \boldsymbol{\theta}) \right] \right] \quad (27)$$

Next, we need to show the boundedness of the gradient estimator variance. Let

$$G_{\mathcal{B}} = \frac{1}{|\mathcal{B}|} \sum_{(x_i, y_i) \in \mathcal{B}} \left[ \nabla_{\boldsymbol{\theta}} \ell(\hat{y}_{\boldsymbol{\theta}}(\mathbf{x}_i), y_i) + \lambda \sum_{\substack{j \in \mathcal{Y}, \\ k \in \mathcal{S}}} \left[ A_{jk} \nabla_{\boldsymbol{\theta}} F_j(\mathbf{x}_i; \boldsymbol{\theta}) \mathbb{1}(s_i = k) - f^*(A_{jk}) \pi_k \nabla_{\boldsymbol{\theta}} F_j(\mathbf{x}_i; \boldsymbol{\theta}) \right] \right]$$

We need to show for a given data batch:

$$\mathbb{E}[\|G_{\mathcal{B}} - G_n\|_2^2]$$

where $G_n$ is the gradient with respect to all $n$ data points (when $\mathcal{B} = \{1, \dots, n\}$. Note that:

$$\|G_{\mathcal{B}} - G_n\|_2^2 \leq 2\|G_{\mathcal{B}}\|_2^2 + \|G_n\|_2^2$$

Thus, it suffices to show that the gradient is bounded for any given $\mathcal{B}$ batch. Since the samples are independent of each other and identically distributed from $\mathbb{P}_{\text{train}}$ (IID samples), the second-order moment of the average over $|\mathcal{B}|$ data points is $1/|\mathcal{B}|$ times the variance of a single data point.

Thus, we need to show the boundedness of the gradient for a given data point drawn from the training distribution:

$$\left[ \nabla_{\boldsymbol{\theta}} \ell(\hat{y}_{\boldsymbol{\theta}}(\mathbf{x}), y_i) + \lambda \sum_{\substack{j \in \mathcal{Y}, \\ k \in \mathcal{S}}} \left[ A_{jk} \nabla_{\boldsymbol{\theta}} F_j(\mathbf{x}_i; \boldsymbol{\theta}) \mathbb{1}(s_i = k) - f^*(A_{jk}) \pi_k \nabla_{\boldsymbol{\theta}} F_j(\mathbf{x}_i; \boldsymbol{\theta}) \right] \right] \quad (28)$$

Based on the first assumption:

$$\|\nabla_{\boldsymbol{\theta}} \ell(\hat{y}_{\boldsymbol{\theta}}(\mathbf{x}), y_i)\|_2 \leq G \quad (29)$$

Based on the second assumption:

$$\|A_{jk} \nabla_{\boldsymbol{\theta}} F_j(\mathbf{x}_i; \boldsymbol{\theta}) \mathbb{1}(s_i = k)\|_2 \leq L A_{jk} \quad (30)$$

$$\|\pi_k f^*(A_{jk}) \nabla_{\boldsymbol{\theta}} F_j(\mathbf{x}_i; \boldsymbol{\theta})\|_2 \leq \pi_k L f^*(A_{jk}) \quad (31)$$

These terms are bounded if $A_{jk}$ is bounded and $f^*(A_{jk})$ is bounded for any $A_{jk}$. This holds true for all $f$-divergences given assumptions 3 and 4. To see why, it suffices to find the optimal solution of each $f$-divergence by setting the gradient zero with respect to $A_{jk}$. In all cases, the solution is a combination of $\mathbb{P}_{s_k}$ and $\mathbb{P}_{\hat{y}_j}$ terms that are non-zero and bounded (by assumptions 3 and 4). Since each term is bounded in (28), the expectation of the squared norm is also bounded, according to Lemma F.6.

Finally, given that the estimator is unbiased, and the variance is bounded (Assumption 4.1 holds in Lin et al. (2020)), the two-time-scale stochastic gradient descent-ascent algorithm (which is Algorithm 1) finds an $\epsilon$-stationary solution of the Problem in $\mathcal{O}(\frac{1}{\epsilon^8})$ according to Theorem 4.9 in Lin et al. (2020). □

*Remark* F.7. For the case of strongly convex $f$-divergence (e.g. $\chi^2$ divergence), the convergence rate of $\mathcal{O}(\kappa^3 \epsilon^{-4})$ can be obtained (Theorem 4.5 in Lin et al. (2020)). Such an iteration complexity holds for the batch size of $\mathcal{O}(\frac{\kappa \sigma^2}{\epsilon^2})$. If the batch size is as small as one, the rate will be $\mathcal{O}(\kappa^3 \epsilon^{-5})$.

*Remark* F.8. If the batch size is $n$ (deterministic), a rate of $\mathcal{O}(\epsilon^{-6})$ can be obtained according to Theorem 4.8 in Lin et al. (2020). Note that it does not necessarily translate to a better runtime than the stochastic case. Because the per iteration cost of evaluating the gradient for $n$ data points can be much higher than evaluating on just 1 (or a small number of) data points.

# G  A FASTER (BUT DOUBLE-LOOP) FIRST-ORDER OPTIMIZATION ALGORITHM FOR OPTIMIZING ($f$-FERM)

We apply SREDA (Luo et al., 2020) to find an $\epsilon$ stationary solution of Problem ($f$-FERM). Note that SREDA works for non-convex strongly-concave min-max problems. We can directly apply the algorithm when $f$ is the $\chi^2$-divergence. In the cases that the function is concave but not strongly concave (e.g., KL divergence and Reverse KL), we first consider the following approximation:

$$\min_{\boldsymbol{\theta}} \max_{\mathbf{A}} \quad \frac{1}{n} \sum_{i=1}^{n} \left[ \ell(\hat{y}_{\boldsymbol{\theta}}(\mathbf{x}_i), y_i) + \lambda \sum_{\substack{j \in \mathcal{Y}, \\ k \in \mathcal{S}}} \left[ A_{jk} F_j(\mathbf{x}_i; \boldsymbol{\theta}) \mathbb{1}(s_i = k) - f^*(A_{jk}) \pi_k F_j(\mathbf{x}_i; \boldsymbol{\theta}) - \frac{\epsilon}{2} \|A_{jk}\|^2 \right] \right]$$
(32)

The maximization problem is an $\epsilon$-strongly concave problem. If we apply SREDA (see Algorithm 3 in Luo et al. (2020)), we find an $\epsilon$ stationary solution of Problem (32) in $\mathcal{O}(\kappa^3 \epsilon^{-3})$ where $\kappa = \frac{L}{\mu}$ is the condition number. In our case, $\mu$, the strong concavity modulus can be set to the desired accuracy $\epsilon$ so that solving the approximate problem (32) leads to an approximate stationary point of the original problem. Therefore, the rate of convergence will be $\mathcal{O}(\epsilon^{-6})$. Note that applying SREDA finds an $\epsilon$ stationary solution of Problem (32). Similar to Theorem 3.1, since the added regularization term is small enough, the obtained solution is a $\mathcal{O}(\epsilon)$-stationary solution of the original problem ($f$-FERM). An important note is that the SREDA algorithm (line 13 in Algorithm 3 (Luo et al., 2020)) has a nested loop compared to the SGDA algorithm proposed in Algorithm 1. Therefore, the $\mathcal{O}(\epsilon^{-6})$ iteration complexity bound does not necessarily translate to an improved runtime in practice. Algorithm 2 describes SREDA applied to Problem (32). For the simplicity of the presentation, define the summation argument over $n$ data points as $\phi(\mathbf{x}_i, y_i, s_i, \boldsymbol{\theta}, \mathbf{A})$. The ConcaveMaximizer module is

---

**Algorithm 2** SREDA Algorithm For Solving ($f$-FERM)

---

1: **Input**: periods $q$, m > 0, Number of iterations T, step-size $\eta_{\boldsymbol{\theta}}$, fairness parameter $\lambda \geq 0$, iteration number $T$, Batchsizes $S$ and $R$.
2: **for** $t = 1, \ldots, T$ **do**
3:    **if** $t \bmod q = 0$ **then**
4:       Draw $S$ samples $(\mathbf{x}_1', y_1'), \ldots, (\mathbf{x}_S', y_S')$
5:       $\mathbf{v}_t = \frac{1}{S} \sum_{i=1}^{S} \nabla_{\boldsymbol{\theta}} \phi(\mathbf{x}_i, y_i, s_i, \boldsymbol{\theta}, \mathbf{A})$
6:       $\mathbf{u}_t = \frac{1}{S} \sum_{i=1}^{S} \nabla_{\mathbf{A}} \phi(\mathbf{x}_i, y_i, s_i, \boldsymbol{\theta}, \mathbf{A})$
7:    **else**
8:       $\mathbf{v}_t = \mathbf{v}_t'$
9:       $\mathbf{u}_t = \mathbf{u}_t'$
10:    $\boldsymbol{\theta}_{t+1} = \boldsymbol{\theta}_t - \eta_{\boldsymbol{\theta}} \mathbf{v}_k$
11:    $(\mathbf{A}_{t+1}, \mathbf{v}_{t+1}', \mathbf{u}_{t+1}') = \text{ConcaveMaximizer}(t, m, R, \boldsymbol{\theta}_t, \boldsymbol{\theta}_{t+1}, \mathbf{A}_t, \mathbf{u}_t, \mathbf{v}_t)$
12: **Return:** $\theta^T$

---

described in Algorithm 4 in Luo et al. (2020).

# H  A FIRST-ORDER OPTIMIZATION ALGORITHM FOR OPTIMIZING (10)

This section presents a first-order optimization algorithm for optimizing (10). The details are presented in Algorithm 3. Further, we show the convergence of the algorithm to an $\epsilon$-stationary solution of Problem (10) in $\mathcal{O}(\epsilon^{-8})$.

**Theorem H.1.** *Assume that $\ell(\cdot, \cdot)$ and $\mathcal{F}_j(\cdot, \boldsymbol{\theta})$ are Lipschitz continuous for any given $j$ and $\boldsymbol{\theta}$ and their gradients are L-Lipshitz. Further, assume that $\mathbb{P}(s = k) > 0$ for all protected groups and $\mathbb{P}(\hat{y}_{\boldsymbol{\theta}} = j) > 0$ at every iteration for all labels $j$. Then, for any given batch size $1 \leq |\mathcal{B}| \leq n$, Algorithm 3 finds an $\epsilon$-stationary solution of ($f$-FERM) in $\mathcal{O}(\frac{1}{\epsilon^8})$ for any given $\epsilon > 0$.*

The proof of the theorem is similar to Theorem 2.2 as the objective function is non-convex concave.

One can obtain faster algorithms under additional assumptions. For example, if the set for $\theta$ is assumed to be compact (e.g., we restrict the norm of the weight of the gradient), then we can accelerate the

algorithm to $O(\epsilon^{-6})$, see Rafique et al. (2022). Moreover, if we consider full batch sizes, we can utilize Algorithm 2 in Ostrovskii et al. (2021a). This will give you the rate of convergence of $O(\epsilon^{-2})$ (Theorem 5.2).

---

**Algorithm 3** Gradient-Regularization Robust Training algorithm

---

1: **Input**: $\boldsymbol{\theta}^0 \in \mathbb{R}^{d_\theta}$, step-sizes $\eta_{\boldsymbol{\theta}}, \eta_\alpha$, fairness parameter $\lambda \geq 0$, iteration number $T$, Batchsize $[b]_t$
2: **for** $t = 1, \ldots, T$ **do**
3:     Sample minibatch of data $\mathbf{b}_t = \{(\mathbf{x}_1, \mathbf{y}_1), \cdots, (\mathbf{x}_b, \mathbf{y}_b)\}$
4:     Estimate $\mathbb{P}(\hat{\mathbf{y}}_{\boldsymbol{\theta}^t})$ for minibatch $\mathbf{b}_t$
5:     **repeat**
6:         $dA_{jk} = \nabla_{\mathbf{A}}(A_{jk}\mathbb{P}_{\hat{\mathbf{y}}_{\boldsymbol{\theta}},\mathbf{s}} - f^*(A_{jk})\mathbb{P}_{\hat{\mathbf{y}}_{\boldsymbol{\theta}}}\mathbb{P}_{\mathbf{s}})$
7:         $A_{jk} = A_{jk} + \eta_\alpha \, dA_{jk}$
8:     **until** Convergence to $A_{jk}^*$
9:     Obtain closed form expressions: $\frac{\partial}{\partial \boldsymbol{\theta}}||\nabla_{\mathbb{P}}\mathcal{D}_f(\mathbb{P}||\mathbb{Q})||_2^2$ and $\frac{\partial}{\partial \boldsymbol{\theta}}||\nabla_{\mathbb{P}}\mathcal{D}_f(\mathbb{P}||\mathbb{Q})||_2^2$ in terms of $\mathbb{P}_{\hat{\mathbf{y}}_{\boldsymbol{\theta}}}$
10:     $d\boldsymbol{\theta} = \nabla_{\boldsymbol{\theta}}\Big[\ell(\boldsymbol{\theta}^{t-1}, \mathbf{x}, \mathbf{y}) + \lambda\Big[\mathcal{D}_f(\hat{\mathbb{P}}||\hat{\mathbb{Q}}) + \epsilon\Big(||\nabla_{\mathbb{P}}\mathcal{D}_f(\hat{\mathbb{P}}||\hat{\mathbb{Q}})||_2^2 + ||\nabla_{\mathbb{Q}}\mathcal{D}_f(\hat{\mathbb{P}}||\hat{\mathbb{Q}})||_2^2\Big)\Big]\Big]$
11:     $\boldsymbol{\theta}^t = \boldsymbol{\theta}^{t-1} - \eta_{\boldsymbol{\theta}} \, d\boldsymbol{\theta}$
12: **Return:** $\boldsymbol{\theta}^T$

---

## I   PROOF OF EQUATION (12)

To show Problem (8) is equivalent to (12) under $\ell_p$ norm and the probability simplex constraint relaxation, note that the maximization problem in (8) is a constrained convex maximization with respect to $\mathbb{P}$. Therefore, there is a global solution on the boundary. The maximum problem under $\ell_\infty$ can be written as:

$$\max_{\substack{||\mathbb{P}-\hat{\mathbb{P}}||_\infty \leq \delta \\ ||\mathbb{Q}-\hat{\mathbb{Q}}||_\infty \leq \delta}} \mathcal{D}_f(\mathbb{P}||\mathbb{Q}), \tag{33}$$

or equivalently:

$$\max_{\substack{||\mathbb{P}-\hat{\mathbb{P}}||_\infty \leq \delta \\ ||\mathbb{Q}-\hat{\mathbb{Q}}||_\infty \leq \delta}} \sum_{j=1}^m \mathbb{Q}_j f\Big(\frac{\mathbb{P}_j}{\mathbb{Q}_j}\Big), \tag{34}$$

For the choice of KL-divergence ($f(t) = t\log(t)$) and $\chi^2$ divergence ($f(t) = (t-1)^2$), $f$ is a non-decreasing function. Fixing a $j \in \{1, \ldots, m\}$, the maximum with respect to is $\mathbb{P}_j$ attained when $\mathbb{P}_j$ is maximized. The maximum of $\mathbb{P}_j$ is obtained on the boundary where $\delta$ is added to $\hat{\mathbb{P}}_j$. Since $\hat{\mathbb{P}}_j + \delta$ should be a probability value, if it is larger than 1, we project it back to 1. As a result, the maximum of $\mathbb{P}_j$ is $\max(\hat{\mathbb{P}}_j + \delta, 1)$. Further, $f$ in both choices of $f$ is super-linear, meaning that $\mathbb{Q}_j f\Big(\frac{\mathbb{P}_j}{\mathbb{Q}_j}\Big)$ is a non-increasing function with respect to $\mathbb{Q}_j$. Thus, its maximum with respect to $\mathbb{Q}_j$ is attained when $\mathbb{Q}_j$ is minimized. Therefore, the optimal solution is either $\hat{\mathbb{Q}}_j - \delta$, or if it goes less than 0, we project it back to 0. Applying the same argument to all $j$'s, we obtain Equation (12).

## J   DETAILS OF TUNING HYPERPARAMETERS

In all experiments, we set $\eta_{\boldsymbol{\theta}} = 10^{-5}$ and $\eta_\alpha = 10^{-6}$. Further, we train the model with $\lambda = 0$ for 300 epochs, and then we set $\lambda$ to the considered value. We continue the training until 2000 epochs. The range of $\lambda$ to get each point in the tradeoff figures is varied for different $f$-divergences. The KL-divergence $\lambda$ range is $[0, 150]$. For $\chi^2$ divergence it is $[0, 300]$ and for the reverse KL it is $[0, 50]$. Moreover, the $\lambda$ range for JS and Squared Hellinger is $[0, 110]$ and $[0, 250]$. Note that larger values outside the range lead to models with 0 predictions for all values.

In the DRO case, aside from $\lambda$ we must tune $\epsilon$, the robustness parameter. To achieve the best result, we have two different strategies depending on the availability of the data from the target domain. Suppose we have access to a collection of data points from the target domain. In that case, we consider it as the validation set to choose the optimal combination of $\lambda \in \{0.1, 0.5, 1, 2, 5, 10, 20, 50\}$ and $\delta \in \{0.01, 0.02, 0.05, 0.1, 0.2, 0.5, 1, 2, 5, 10\}$. In the second scenario, when we do not have any

access to target domain data, we perform a $k$-fold cross-validation on the source data. A more elegant way is to create the validation dataset by oversampling the minority groups. Having access to the oversampled validation set, we choose the optimal $\lambda$ and $\delta$ similar to the first scenario. In the experiment regarding Figure 4, we reserve $5\%$ of data from the target domain for validation (scenario 1). In Figure 2, we apply scenario 2 to tune the hyperparameters $\lambda$ and $\delta$.

## K    FURTHER EXPERIMENTS ON OTHER DATASETS AND NOTIONS OF FAIRNESS

In this section, we perform ($f$-FERM), (Hardt et al., 2016), (Mary et al., 2019), and (Baharlouei et al., 2020) to COMPAS [2] and German Credit datasets [3]. In the experiment on COMPAS, we use equality of opportunity as the measure of fairness violation, while in the German Credit dataset experiment, we use equalized odds. The results show that ($f$-FERM) is significantly better than other approaches regarding the accuracy-fairness tradeoff. The batch size is equal to $64$ for all methods.

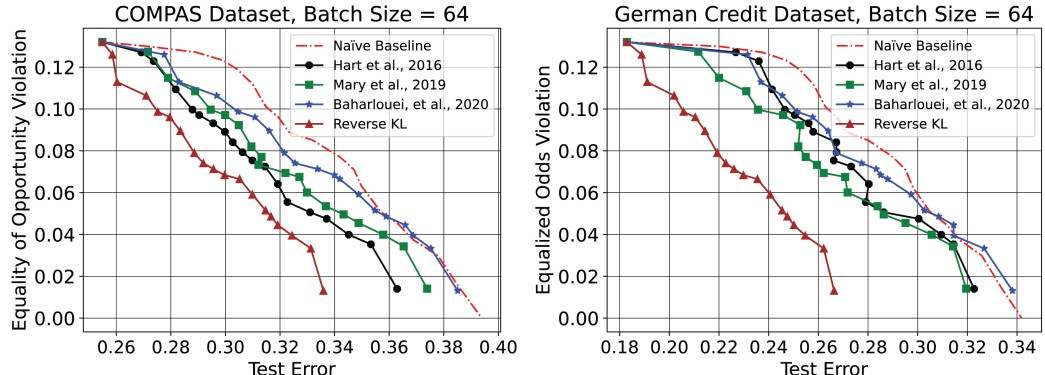

Figure 5: Performance of the trained fair models on COMPAS and German Credit Datasets.

---

[2]https://www.kaggle.com/datasets/danofer/compass
[3]https://archive.ics.uci.edu/dataset/144/statlog+german+credit+data