# OpenReview forum: "f-FERM: A  Scalable Framework for  Robust Fair Empirical Risk Minimization"
_ICLR.cc/2024/Conference — ICLR 2024 poster_

### Official Review · Reviewer_hKcK · 2023-10-22

**Soundness:** 3 good
**Presentation:** 3 good
**Contribution:** 2 fair
**Rating:** 5
**Confidence:** 3

**Summary:**

This paper proposes a new framework for fairness, based on f-divergence and mutual information. This method is called f-FERM. From theory of minimax optimization, it shows the convergence to the optimum of f-FERM. Empirically, f-FERM also shows improvement over existing baselines.

--post rebuttal--
Thank you for the rebuttal and my understanding of this paper got sharpened. I think the fairness definition needs to be clearly stated in the paper, and the uncertainty of which f-divergence seems a bit confusing to me. Unfortunately I cannot change the score based on my point of view.

**Strengths:**

1. The method is quite straightforward. Suppose the output $y$ and the sensitive attribute $s$ are independent, then the f-Mutual Information term becomes zero. Therefore, the $f$-FERM encourages independence and thus fairness.
2. Using variational bound and convergence theory of minimax, it proves the convergence of SGDA to a stationary point.
3. Experimental results show the improvement of $f$-FERM over existing benchmarks.

**Weaknesses:**

1. This paper is missing a related reference called TERM (Tilted Empirical Risk Minimization) which also studies fairness.
2. The main formulation (1) could be simplified. The f-divergence between the joint distribution and the product of marginals is called f-mutual information. See https://openreview.net/forum?id=ZD03VUZmRx for example and the references therein.
3. SGDA for minimax optimization is highly inefficient and Theorem 2.2 has a really weak convergence result. How is it related to the experiments?
4. The paper didn't discuss which $f$ (in f-div) is the best choice in detail.
5. All the experiments are conducted on small-scale datasets.
6. The definition of fairness is not clear to me. This paper considers both DP and EO, but which one is what f-FERM is trying to approach?

**Questions:**

See above.

---

> ### Author Response · Authors · 2023-11-23
> **Response to Reviewer hKcK**
>
> We appreciate the time you have taken to read our manuscript and provide feedback to us. Below is our point-to-point response to your comments.
>
> > Missing a related reference called TERM.
>
> This paper was cited and discussed in the original submission. The proposed fairness notion in that paper is **based on equal performance for different sub-groups**, which is different from our fairness notions. In contrast to our work, **their paper focus is not on finding convergent stochastic algorithms or distributionally robust fair learning**.
>
> > The main formulation could be simplified. The f-divergence between the joint distribution and the product of marginals is f-mutual information (https://openreview.net/forum?id=ZD03VUZmRx)
>
> Thank you for bringing up this work, which we cited in our revision. While f-mutual information is proposed in the contrastive learning context in the paper you mentioned, **their first formulation is exactly equal to our regularization term (for the case of demographic parity).** Therefore, though we do not use the same terminology for our regularizer, they are equivalent, and none is simpler than the other. The other developments in the paper are based on assuming the joint feature distribution is proportional to the Gaussian kernel, which we do not rely on. As we explained in our general response, our contribution is the reformulation leading to a stochastic optimization algorithm and its extension to the distributionally robust settings.
>
> > SGDA for minmax optimization is inefficient. Theorem 2.2 has a weak convergence result. How is it related to experiments?
>
> Good question! We agree that SGDA can be improved under some additional assumptions (see our general response, the added discussion in the paper after Theorem 2.2, and Appendix G in the revised manuscript). However, SGDA is widely used for minmax optimization due to its simplicity and ease of tuning. There are **other double-loop and triple-loop algorithms with overall less iteration complexity, as we included in the paper**. However, **the underlying condition for using all these algorithms is access to an unbiased gradient estimator with a bounded variance.** This condition is satisfied in our formulation and will allow users to choose their own favorite algorithm. In our experiments, we found SGDA dependable because of its **single-loop nature** and the **small number of parameters for tuning.**
>
> > All the experiments are on small-scale datasets.
>
> We have used all popular benchmark datasets, including Adult, German Credit, and COMPAS datasets. For the DRO section, there are indeed not many available real-world datasets containing natural distribution shifts. The New Adult dataset (which we used in our experiments) is one of the newest and largest datasets available (50 states, with each having its own distribution). We believe our experiments on a wide range of datasets are the reason that some other reviewers found our experiments extensive and comprehensive. Having said that, we would love to include additional experiments in the GitHub repository of our paper if you have any specific dataset suggestions.
>
> > The paper didn't discuss which f  is the best choice in detail.
>
> This is an important and natural question. As can be seen in our experiments, **there is no single divergence that is always better than others in all settings.** In the case of binary-sensitive attributes and full batch algorithms, their performance is almost the same. In the case of non-binary sensitive attributes, if the batch size is small, we observe that KL divergence works impressively well (we mentioned this in the analysis of Figure 1). We believe in applications, the choice of the $f$-divergence can be viewed as a hyperparameter that can be tuned by cross-validation. We have **clarified this point in our revised manuscript at the beginning of section 4**. An interesting theoretical future research direction could be why and when each of these $f$ divergence measures can outperform the others.
>
> > The definition of fairness is not clear to me. This paper considers both DP and EO, but which one is what f-FERM approach?
>
> Our formulation can handle **any group fairness measure that is based on the [conditional] independence between sensitive attributes** and predictions. This fact is explained at the beginning of section 2. Specifically, our framework can handle demographic parity, equalized odds, and equality of opportunity. In the main body, we have focused on demographic parity. However, by a simple conditioning of the probability measures, we can generalize to other notions. For example, if instead of $s$ and $\hat{y}$, we consider $s | y = 1$, and $\hat{y} | y = 1$ in (6), then we obtain f-ferm for equality of opportunity. Please see the discussion in Appendix A.
>
> We appreciate your time and feedback, and we hope our response addresses your concerns. If so, we would appreciate it if you could reflect it in your evaluation of our paper.

---

### Official Review · Reviewer_zLcF · 2023-10-24

**Soundness:** 3 good
**Presentation:** 3 good
**Contribution:** 3 good
**Rating:** 6
**Confidence:** 4

**Summary:**

The paper proposed a unified stochastic optimization framework for fair ERM based on f-divergence measures.  The main idea is to reformulate it as a minimax problem using the Legendre-Fenchel conjugate function. This minimax formulation facilitates the development of the standard stochastic gradient descent and ascent algorithms for solving fair ERM. The convergence guarantees are stated in Theorem 2.2 which is mainly a corollary of the results in Lin et al. (2020) since this formulation is concave w.r.t. the dual variable.   The paper also addressed the problem of distribution shift and reformulated it using Lagrangian relaxation as a non-convex and non-concave problem. Extensive experiments are conducted to validate the proposed algorithms.

**Strengths:**

1. A unified formulation for fair ERM using the f-divergence and then minimax reformulation which facilitates the application of SGDA.
2.  A robust variant to address the distribution shift
3. Extensive  and convincing experiments

**Weaknesses:**

1. The proposed unified formulation and its minimax reformulation seem incremental as similar minimax forms have appeared in many existing works of fair machine learning
2. The convergence analysis is straightforward from the paper by Lin et al. (2020).

**Questions:**

NA

---

> ### Author Response · Authors · 2023-11-23
> **Response to Reviewer zLcF**
>
> We are delighted that your characterization of the strength of our paper aligns with ours. Particularly, we are encouraged that you found our unified framework, algorithm, and robustified versions solid contributions and our numerical experiments extensive and convincing. We appreciate the time you spent reading our paper and providing us feedback to improve our manuscript. Below is our point-to-point response to your concerns:
>
>
> > The proposed unified formulation and its minimax reformulation seem incremental, as similar minimax forms have appeared in many existing works of fair machine learning
>
> You are correct that there exist other min-max formulations in the literature, such as Renyi Fair Inference (Baharlouei et al., 2020) and Fairness-Aware Learning (https://proceedings.mlr.press/v97/mary19a/mary19a.pdf). However, the algorithms they provided only work for **full-batch** settings.  The reason is that it is not straightforward for their formulation to derive an unbiased estimator of the gradient (as a necessary condition for SGDA and other first-order algorithms) for small batches. In addition, there are two other aspects that completely distinguish our work from existing works. First, our framework covers a **wide range of divergences (and it is not a single divergence)**. This will give the practitioner the option to use the most suitable/favorite divergence given their application. Second, our framework **can handle settings involving distribution shifts**. We believe this step is another major contribution of our work which sets us apart from previous works and should not be overlooked.
>
> > The convergence analysis is straightforward from the paper by Lin et al. (2020).
>
> We agree that our main contribution is not the convergence analysis of the algorithm. Our primary contribution is that **we provide a formulation leading to an unbiased estimator of the gradient with a bounded variance** that can be naturally obtained. Having access to such an estimator enables us to use existing first-order stochastic optimization methods with convergence guarantees to solve the problem. Indeed, **for both the iid setting and the distribution shift setting**, our novel formulation can be solved with existing optimization algorithms. To clarify this point, we added a discussion in our revised paper explaining existing algorithms that can be used in our problem. Please see the paragraph after Theorem 2.2 and the new Appendix G.
> \
> \
> Thank you for your time and for your valuable feedback. We hope our response adequately clarifies our contributions, and we would be greatly thankful if you could consider this in your evaluation of our paper.

---

### Official Review · Reviewer_PRWu · 2023-10-28

**Soundness:** 4 excellent
**Presentation:** 3 good
**Contribution:** 4 excellent
**Rating:** 8
**Confidence:** 3

**Summary:**

Fairness constraints, such as DP and EO are hard to optimise directly thanks to their discrete nature. Existing bias mitigation approaches propose various differentiable proxies, typically correlation based, such as Zafar et al. They however do not allow unbiased small-batch estimation of the corresponding gradient.

The present paper uses f-divergence as a proxy for independence (they include EO & DP). They introduce a gradient based method, where that has unbiased estimate and therefore allows arbitrary small batch size. In the experiments they use batch size as small as 8. Arguably, this can allow to scale the method to larger problems.

To my knowledge, this paper presents the first theoretical result for gradient based optimisation of fair classifiers.

I want to ask authors to clarify, how do they obtain classifier $ \hat{y}$?
Typically, one obtains differentiable probabilities $p_{\theta}(y = j|x)$, and then maximising the probability yields the classification. However, the expression (4) suggest that the estimator is sampled.
This may harm the accuracy in practice. It is ok to use $F_j$ as a proxy, but it is better if you clarify that explicitly for applied people who may potentially use your method.

Secondly, in Theorem 2.2 the stated convergence time is $O(\epsilon^{-8})$ which seems rather slow. Does it affect the amount of epochs you use in the experiments, is it unusually large?

Thirdly, could you please clarify directly in Proposition 2.2. what is the dimension of variables $A_{jk}$

**Strengths:**

SGD with unbiased gradient updates for fair classification.

Additional result for robust fair classification.

Theoretical result for SGD for fair classification.

**Weaknesses:**

mentioned in summary

**Questions:**

mentioned in summary

---

> ### Author Response · Authors · 2023-11-23
> **Response to Reviewer PRWu**
>
> We are encouraged that you found our results novel and interesting and that you understood our main contributions. We would also like to thank you for your constructive feedback. Below, please find our point-to-point response to your comments:
>
> > I want to ask authors to clarify, how do they obtain classifier $\hat{y}$? Typically, one obtains differentiable probabilities , and then maximising the probability $p_{\theta}(y=j|x)$ yields the classification. However, the expression (4) suggest that the estimator is sampled. This may harm the accuracy in practice. It is ok to use $F_j$ as a proxy, but it is better if you clarify that explicitly for applied people who may potentially use your method.
>
> This is a great point. As you correctly pointed out, our formulation is based on the randomized prediction rule where the label is samples. More specifically, $F_i(x)$ represents the probability of predicting class $i$ for data point $x$. However, as you correctly mentioned, this could harm the accuracy in practice. Thus, in our numerical experiments, we used the standard **maximum likelihood decoding** based on the output of the softmax layer, i.e., the predicted label is the label with the highest logit value. We have clarified this point in our revision, see page 8 in the revised manuscript.
>
> > Secondly, in Theorem 2.2 the stated convergence time is $O(\epsilon^{-8})$ which seems rather slow. Does it affect the amount of epochs you use in the experiments, is it unusually large?
>
> The rate of $O(\epsilon^{-8})$ holds for all the f-divergences and all batch sizes for SGDA algorithm. However, under certain additional conditions or for special choices of f-divergence, this rate can be improved. Please see our general response and the added Appendices in the revised manuscript.
>
> > Thirdly, could you please clarify directly in Proposition 2.2. what is the dimension of variables $A_{jk}$?
>
> $\mathbf{A}$ is a $|Y| \times |S|$ matrix where **$Y$ is the set of possible labels, and $S$ is the set of sensitive attribute values**. $A_{jk}$ represents the entry of matrix A corresponding to the label j and sensitive attribute value k. Therefore, $A_{jk}$ is a **scalar**. We believe that our use of bold font for that led to the confusion. Hence in our rebuttal, we changed it to regular font.
> \
> \
> \
> Thank you again for your detailed and constructive comments. We hope our response answers your question. Particularly, your comments on the randomized nature of our classifier and the rate of convergence of the algorithm are important and we hope our changes in the paper make it clear for future readers.

---

### Official Review · Reviewer_o4nf · 2023-11-02

**Soundness:** 3 good
**Presentation:** 3 good
**Contribution:** 2 fair
**Rating:** 5
**Confidence:** 4

**Summary:**

This paper first presents a framework for fair ERM using f-divergence. Next, the authors propose a distributionally robust fair ERM framework, where the ambiguity set is constructed using $\ell_p$-norm. For the theoretical part, a convergence analysis of the stochastic optimization for the ERM formulation is proposed, leveraging the results from Lin et al. (2020). The authors perform comprehensive numerical study to validate the superior performance of their proposed framework.

**Strengths:**

- The formulation, optimization algorithm, and convergence analysis for the ERM formulation seem correct and novel.
- The numerical study is comprehensive to demonstrate the effectiveness of their proposed algorithm.

**Weaknesses:**

- The only novel theoretical part is the convergence analysis of the SGDA for solving the ERM formulation. The corresponding sample complexity for finding $\epsilon$-statariony solution of (f-FERM) is $O(\epsilon^{-8})$. However, I wondered if the authors use the state-of-the-art optimization algorithm for solving nonconvex-concave min-max games and if the sample complexity is currently the optimal one in the literature. If so, the authors should highlight it in the literature.
- For the distributionally robust formulation (8), the authors used $\ell_p$-norm to model the ambiguity set, but I am afraid that this usage may not be a novel choice because it is not a flexible choice as Wasserstein, $f$-divergece, or MMD, to quantify the difference between distributions. I encourage the authors consider the extension to other choices of ambiguity sets.
- When the ambiguity set of formulation (8) is small, the authors propose a first-order approximation formula to solve it (see Eq.(10)). Although the authors present the approximation error between formula (8) and (10), the complexity for solving Eq.(10) is not presented in this paper.
- When the ambiguity set of formulation (8) is potentially large, the first-order approximation formula may not achieve good performance. To this end, the authors consider another formulation in Section 3.2. However, I have several concerns of the deviation:
  * Can the authors add more details regarding the sentence "One can easily see that the optimal $p_j=\min(\hat{p}_j+\delta, 1)$ and $q_j=\max(\hat{q}_j-\delta,0)$"? It is not obvious for readers to check this point.
  * What is the meaning of the scalar $\delta>0$? It is not introduced. if I remember correctly, in standard f-divergence DRO it is the Lagrangian multiplier corresponding to the probability simplex constraint that needs to be optimized.

**Questions:**

See the weakness part.

---

> ### Author Response · Authors · 2023-11-23
> **Response To Reviewer o4nf Part 1/2**
>
> We appreciate your time and valuable feedback. We are glad that you found the formulation novel, the theoretical analysis sound, and the experiments comprehensive. Below, please find our point-to-point response to your comments:
>
> > The only novel theoretical part is the convergence analysis of the SGDA for solving the ERM formulation. The corresponding sample complexity for finding -the statariony solution of (f-FERM) is $O(\epsilon^{-8})$. However, I wondered if the authors use the state-of-the-art optimization algorithm for solving nonconvex-concave min-max games and if the sample complexity is currently the optimal one in the literature. If so, the authors should highlight it in the literature.
>
> We agree with you that faster algorithms can be used to solve our problem (under some additional assumptions). As explained in our general response above, the rates can be improved under certain additional assumptions and via using more sophisticated algorithms in the literature. However, we chose the simple SGDA algorithm due to its simple implementation (single loop with much fewer tuning parameters compared to other existing methods). In addition, SGDA may have generalizability benefits as pointed out in the general response. Nevertheless, this is not the major contribution of our work. We clarified these points in our revised manuscript. However, for the sake of completeness and **based on your suggestion, we added a detailed and comprehensive discussion on what rates can be achieved using existing algorithms on page 5, Appendix F, and Appendix G**. Having said that, we do think that most of these algorithms are **less practical than SGDA** due to their heavy hyperparameter tuning needs. This has also been observed in the past for much simpler training tasks (such as regular minimization problems), as SGD is still the workhorse for training convex and nonconvex models, while there are theoretically faster available algorithms out there.
> Regarding the novelty, we agree with your earlier assessment that the formulation of the problem is novel. Our formulation can cover a wide range of divergences and lead to efficient algorithms. Indeed, as explained in our general response, our main contribution to non-robust fair learning is the formulation and the reformulation of it as a **min-max optimization with unbiased (and bounded variance) gradient estimator**. Once we have this step, the rest would mostly rely on existing literature. Similarly, in the robust fair setting, our main contribution is to reformulate the problem in a form that can be efficiently solved. These are non-trivial contributions, in our opinion, and should not be overlooked.
>
> > For the distributionally robust formulation (8), the authors used $\ell_p$-norm to model the ambiguity set, but I am afraid that this usage may not be a novel choice because it is not a flexible choice as Wasserstein, f-divergeces, or MMD, to quantify the difference between distributions. I encourage the authors to consider the extension to other choices of ambiguity sets.
>
> Thank you for your suggestion. This is a theoretically interesting future research direction to pursue. Having said that, we believe this is much less practically relevant in our context. For example, the Wasserstein ball relies on an underlying distance between the random variables. However, the fairness features in our case are **mostly categorical** (e.g., male vs female) and **not easy to define a natural distance on**. More specifically, Wasserstein distance is natural for **continuous variables or ordinal discrete variables** (e.g. movie ratings or age groups). However, it is not natural to consider an order or distance between different races or genders as sensitive attributes. Further, the Fair ERM is studied under the Wasserstein ball in the past for the special case of logistic regression (convex loss) (https://arxiv.org/pdf/2007.09530.pdf). However, even for this simple case, no scalable stochastic algorithm is offered, even in the convex fair ERM. In contrast, our methodology offers scalable algorithms for even nonconvex models. In addition, our extensive experiments show the effectiveness of the proposed method, which is also supported by our theory.
> Regarding the novelty of using $L_p$ ball, to the best of our knowledge, our work **is the first to use $L_p$ uncertainty balls for DRO fair empirical risk minimization.** It is shown to be practically effective in the experiments and is backed by a reasonable amount of theory. Hence, we still believe there is novelty in our work.

---

> ### Author Response · Authors · 2023-11-23
> **Response To Reviewer o4nf Part 2/2**
>
> > When the ambiguity set of formulation (8) is small, the authors propose a first-order approximation formula to solve it (see Eq.(10)). Although the authors present the approximation error between formula (8) and (10), the complexity for solving Eq.(10) is not presented in this paper.
>
> Thanks for bringing up this concern.  At the end of the day, we apply the SGDA algorithm to Problem (10). We showed how to compute an unbiased estimator of the gradient with respect to each set of parameters. The problem is nonconvex-concave, and thus, all the discussions we had in our general response apply here. In particular, the algorithm converges in $O(\epsilon^{-8})$ iterations with a very small batch size requirement (even batch size of $O(1)$). However, one can obtain faster algorithms under additional assumptions. For example, if the set for $\theta$ is assumed to be compact (e.g., we restrict the norm of the weight of the gradient), then we can accelerate the algorithm to $O(\epsilon^{-6})$, see https://arxiv.org/pdf/1810.02060.pdf. Moreover, if we consider full batch sizes, we can utilize Algorithm 2 in the cited work (Ostrovskii et al 2021). This will give you the rate of convergence of $O(\epsilon^{-2})$; see  (Theorem 5.2, Ostrovskii et al 2021). We added these discussions to the paper in Appendix H. Thank you for helping us detect this miscommunicated point and improve the clarity of the paper.
>
> > Can the authors add more details regarding the sentence "One can easily see that the optimal $p_j = \min(\hat{p}_j+\delta, 1)$ and $q_j = \max(\hat{q}_j - \delta, 0)$”. It is not obvious for readers to check this point.
>
> Great point! Since the problem is a constrained convex maximization problem (not minimization), the solution will be on the boundary. Since $f$ is a non-decreasing function for the choice of KL-divergence and chi-square, the maximum value with respect to $P$ occurs when $P$ takes the maximum possible value, which is either $p + \delta$ or $1$. Similarly, we can see that the maximum value with respect to $Q$ happens when $Q$ takes the minimum possible value, which is $q - \delta$ or $0$. We added a detailed step-by-step proof of the claim as an Appendix (Appendix I) to the paper to clarify this point.
>
> > What is the meaning of the scalar 𝛅>0? It is not introduced. if I remember correctly, in standard f-divergence DRO, it is the Lagrangian multiplier corresponding to the probability simplex constraint that needs to be optimized
>
> $\delta$ represents the uncertainty ball radius (it is usually represented by $\epsilon$. Since we already use $\epsilon$ as the optimality approximation, we use $\delta$ to represent the size of the $L_p$ norm uncertainty set. We analyze the DRO problem for small $\delta$ in Section 3.1. Alternatively, when $\delta$ is large (more robustness), we analyze the problem for the special case of $L_{ \infty}$ norm.
> \
> \
> Finally, we would like to thank you for the time you spent reading our paper and providing constructive feedback to us. Your question on the rate of convergence of the algorithm has helped us clarify a main point in our manuscript and improved the overall quality of our paper. We hope our response clarifies our contributions and answers your questions. If so, we would be grateful if you could reflect this in your evaluation score of our paper.

---

### Author Response · Authors · 2023-11-23
**General Response to Reviewers**

We would like to thank the reviewers for their constructive feedback. We are encouraged that you found our formulation, algorithm, and analysis “correct and novel”. We are also delighted that our experiments are found “Comprehensive,” “Extensive,” and “convincing”. While individual responses to each review are provided, we would like to address an important comment about our contribution in section 2 and the convergence rate of Algorithm 1:


The primary contribution of Section 2 is the reformulation of our training problem as a nonconvex-concave min-max optimization so that an unbiased estimator of the gradient with a bounded variance is available. Once this step is done, we can rely on existing works in the literature to obtain convergent algorithms. We used the SGDA algorithm since it is easy to implement (as it is a single-loop algorithm with a small number of hyper-parameters to tune) and is widely used in the machine learning community. In addition, SGDA can lead to better generalizability than some other alternatives (see, e.g., section 4 in https://arxiv.org/pdf/2206.04502.pdf).


The iteration complexity bound provided in Theorem 2.2 is quite general. It holds for all choices of $f$-divergence measures and all batch sizes. It even holds for noncompact (or unbounded) weight parameter domains. However, one can obtain faster convergence rates in special settings and by using more sophisticated algorithms.  For example, we can obtain $O(\epsilon^{-6})$ iteration complexity (Rafique et al. 2018; Zhang et al. 2022). However, such algorithms use nested loops and require more hyperparameter tunings. We provide an example of such faster algorithms in Appendix G. If the $f$-divergence leads to a strongly concave function in $\mathbf{A}$ or satisfies Polyak-Łojasiewicz condition (e.g., for $\chi^2$-divergence), a faster rate of $O(\epsilon^{-5})$ can be obtained for our algorithm (Appendix F). In addition, if a larger batch size of $O(\epsilon^{-2})$ is used, we can further improve this rate to $O(\epsilon^{-4})$ iteration complexity (see Appendix F). Finally, when full batch size is used, then double/triple-loop algorithms can lead to the iteration complexity bounds of $O(\epsilon^{−2})$ in the nonconvex-strongly concave setting and $O(\epsilon^{-3})$ in the general nonconvex-concave setting; see (Kong & Monteiro, 2021; Nouiehed et al., 2019; Ostrovskii et al., 2021b; Thekumparampil et al., 2019). Nevertheless, these are not the main contributions of the paper. As said earlier, our main contribution is our (re-)formulations that lead to efficient stochastic algorithms and their extensions to distributionally robust settings. Also, our framework is a unifying framework covering all $f$-divergence measures. To clarify these points, we added a discussion on page 5 and Appendix F. We also added Appendix G to the paper. Our main changes in the revised manuscript are highlighted in blue.

---

### Meta-Review · Area_Chair_32KY · 2023-12-15

**Metareview:**

This paper proposes a unified formulation of fair empirical risk minimization based on f-divergence measures and shows that it admits a nonconvex-concave minimax optimization reformulation. It then establishes statistical and optimization guarantees for the reformulation. The reviewers generally agree that the problem studied in the paper is interesting, and the results are sufficiently novel. Please address the reviewers' comments in your camera-ready version.

**Justification For Why Not Higher Score:**

While the problem studied in the paper is important and of interest to the community, and the minimax reformulation of the problem is novel, the algorithm used to tackle the problem and its convergence analysis are mostly standard.

**Justification For Why Not Lower Score:**

The reviewers generally agree that the problem studied in the paper is interesting, and the results are sufficiently novel for publication.

---

### Decision · Program_Chairs · 2024-01-16

Accept (poster)